# Optimizing Data Collection for Machine Learning

**Rafid Mahmood**[1]    **James Lucas**[1]    **Jose M. Alvarez**[1]    **Sanja Fidler**[1,2,3]    **Marc T. Law**[1]

[1]NVIDIA    [2]University of Toronto    [3]Vector Institute

{rmahmood, jalucas, josea, sfidler, marcl}@nvidia.com

## Abstract

Modern deep learning systems require huge data sets to achieve impressive performance, but there is little guidance on how much or what kind of data to collect. Over-collecting data incurs unnecessary present costs, while under-collecting may incur future costs and delay workflows. We propose a new paradigm for modeling the data collection workflow as a formal *optimal data collection problem* that allows designers to specify performance targets, collection costs, a time horizon, and penalties for failing to meet the targets. Additionally, this formulation generalizes to tasks requiring multiple data sources, such as labeled and unlabeled data used in semi-supervised learning. To solve our problem, we develop Learn-Optimize-Collect (LOC), which minimizes expected future collection costs. Finally, we numerically compare our framework to the conventional baseline of estimating data requirements by extrapolating from neural scaling laws. We significantly reduce the risks of failing to meet desired performance targets on several classification, segmentation, and detection tasks, while maintaining low total collection costs.

## 1   Introduction

When deploying a deep learning model in an industrial application, designers often mandate that the model must meet a pre-determined baseline performance, such as a target metric over a validation data set. For example, an object detector may require a certain minimum mean average precision before being deployed in a safety-critical setting. One of the most effective ways of meeting target performances is by collecting more training data for a given model.

Determining how much data is needed to meet performance targets can impact costs and development delays. Overestimating the data requirement incurs excess costs from collection, cleaning, and annotation. For instance, annotating segmentation masks for a driving data set takes between 15 to 40 seconds per object. For 100,000 images the annotation could require between 170 and 460 days-equivalent of time [1, 2]. On the other hand, collecting too little data may incur future costs and workflow delays from having to collect more later. For example, in medical imaging applications, this means further clinical data acquisition rounds that require expensive clinician time. In the worst case, designers may even realize that a project is infeasible only after collecting insufficient data.

The growing literature on sample complexity in machine learning has identified neural scaling laws that scale model performance with data set sizes according to power laws [3–10]. For instance, Rosenfeld et al. [6] fit power law functions on the performance statistics of small data sets to extrapolate the learning curve with more data. In contrast, Mahmood et al. [2] consider estimating data requirements and show that even small errors in a power law model of the learning curve can translate to massively over- or underestimating how much data is needed. Beyond this, different data sources have different costs and scale differently with performance [11–14]. For example, although unlabeled data may be easier to collect than labeled data, some semi-supervised learning tasks may need an order of magnitude more unlabeled data to match the performance of a small labeled set. Thus, collecting more data based only on estimation will fail to capture uncertainty and collection costs.

36th Conference on Neural Information Processing Systems (NeurIPS 2022).

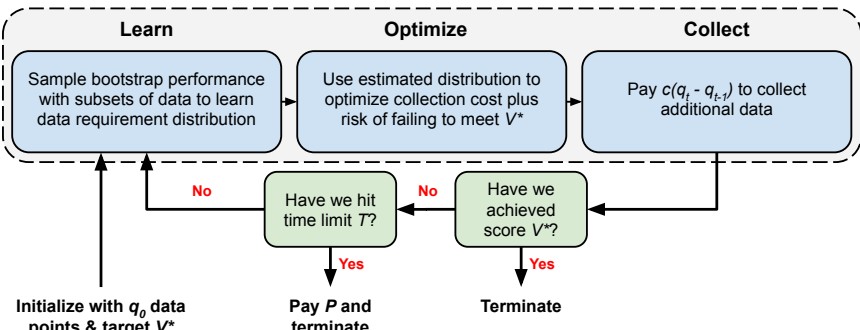

Figure 1: In the optimal data collection problem, we iteratively determine the amount of data that we should have, pay to collect the additional data, and then re-evaluate our model. Our approach, Learn-Optimize-Collect, optimizes for the minimum amount of data $q_t^*$ to collect.

In this paper, we propose a new paradigm for modeling the data collection workflow as an *optimal data collection problem*. Here, a designer must minimize the cost of collecting enough data to obtain a model capable of a desired performance score. They have multiple collection rounds, where after each round, they re-evaluate the model and decide how much more data to order. The data has per-sample costs and moreover, the designer pays a penalty if they fail to meet the target score within a finite horizon. Using this formal framework, we develop an optimization approach for minimizing the expected future collection costs and show that this problem can be optimized in each collection round via gradient descent. Furthermore, our optimization problem immediately generalizes to decisions over multiple data sources (e.g., unlabeled, long-tail, cross-domain, synthetic) that have different costs and impacts on performance. Finally, we demonstrate the value of optimization over naïvely estimating data set requirements (e.g., [2]) for several machine learning tasks and data sets.

Our contributions are as follows. (1) We propose the optimal data collection problem in machine learning, which formalizes data collection workflows. (2) We introduce Learn-Optimize-Collect (LOC), a learning-and-optimizing framework that minimizes future collection costs, can be solved via gradient descent, and has analytic solutions in some settings. (3) We generalize the data collection problem and LOC to a multi-variate setting where different types of data have different costs. To the best of our knowledge, this is the first exploration of data collection with general multiple data sets in machine learning, covering for example, semi-supervised and long-tail learning. (4) We perform experiments over classification, segmentation, and detection tasks to show, on average, approximately a $2\times$ reduction in the chances of failing to meet performance targets, versus estimation baselines.

## 2 Related work

**Neural Scaling Laws.** According to the neural scaling law literature, the performance of a model on a validation set scales with the size of the training data set $q$ via a power law $V \propto \theta_0 q^{\theta_1}$ [5, 6, 8–10, 15–19]. Hestness et al. [5] observe this property over vision, language, and audio tasks, Bahri et al. [9] develop a theoretical relationship under assumptions on over-parametrization and the Lipschitz continuity of the loss, model, and data, and Rosenfeld et al. [6] estimate power laws using smaller data sets and models to extrapolate future performance. Multi-variate scaling laws have also been considered for some specific tasks, for example in transfer learning from synthetic to real data sets [11]. Finally, Mahmood et al. [2] explore data collection by estimating the minimum amount of data needed to meet a given target performance over multiple rounds. Our paper extends these prior studies by developing an optimization problem to minimize the expected total cost of data collected. Specifically, we incorporate the uncertainty in any regression estimate of data requirements and further generalize to multiple data sources with different costs.

**Active Learning.** In active learning, a model sequentially collects data by selecting new subsets of an unlabeled data pool to label under a pre-determined labeling budget that replenishes after each round [20–24]. In contrast, our work focuses on systematically determining an optimal collection budget. After determining how much data to collect, we can use active learning techniques to collect the desired amount of data.

**Statistical Learning Theory.** Theoretical analysis of the sample complexity of machine learning models is typically only tight asymptotically, but some recent work have empirically analyzed these

relationships [25, 26]. Particularly, Bisla et al. [10] study generalization bounds for deep neural networks, provide empirical validation, and suggest using them to estimate data requirements. In contrast, our paper formally explores the consequences of collection costs on data requirements.

**Optimal Experiment Design.** The topic of how to collect data, select samples, and design scientific experiments or controlled trials is well-studied in econometrics [27–29]. For example, Bertsimas et al. [30] optimize the assignment of samples into control and trial groups to minimize inter-group variances. Most recently, Carneiro et al. [31] optimize how many samples and covariates to collect in a statistical experiment by minimizing a treatment effect estimation error or maximizing $t$-test power. However, our focus on industrial machine learning applications differs from experiment design by having target performance metrics and continual rounds of collection and modeling.

## 3   Main Problem

In this section, we give a motivating example before introducing the formal data collection problem. We include a table of notation in Appendix A.

**Motivating Example.** *A startup is developing an object detector for use in autonomous vehicles within the next $T = 5$ years. Their model must achieve a mean Average Precision greater than $V^* = 95\%$ on a pre-determined validation set or else they will lose an expected profit of $P = \$1,000,000$. Collecting training data requires employing drivers to record videos and annotators to label the data, where the marginal cost of obtaining each image is approximately $c = \$1$. In order to manage annual finances, the startup must plan how much data to collect at the beginning of each year.*

Let $z \sim p(z)$ be data drawn from a distribution $p$. For instance, $z := (x, y)$ may correspond to images $x$ and labels $y$. Consider a prediction problem for which we train a model with a data set $\mathcal{D}$ of points sampled from $p(z)$. Let $V(\mathcal{D})$ be a score function evaluating the model trained on $\mathcal{D}$.

**Optimal Data Collection.** We possess an initial data set $\mathcal{D}_{q_0} := \{z_i\}_{i=1}^{q_0}$ of $q_0$ points; we omit the subscript on $\mathcal{D}$ referring to its size when it is obvious. Our problem is defined by a target score $V^* > V(\mathcal{D}_{q_0})$, a cost-per-sample $c$ of collection, a horizon of $T$ rounds, and a penalty $P$ for failure. At the end of each round $t \in \{1, \dots, T\}$, let $q_t$ be the current amount of data collected. Our goal is to minimize the total collection cost while building a model that can achieve the target score:

$$\min_{q_1,\dots,q_T} \ c(q_T - q_0) + P\mathbb{1}\{V(\mathcal{D}_{q_T}) < V^*\} \qquad \text{s.t. } q_0 \le q_1 \le \cdots \le q_T$$

$$= \min_{q_1,\dots,q_T} \ c\sum_{t=1}^{T}(q_t - q_{t-1}) + P\mathbb{1}\{V(\mathcal{D}_{q_T}) < V^*\} \qquad \text{s.t. } q_0 \le q_1 \le \cdots \le q_T \tag{1}$$

The collection cost is measured by the difference in data set size between the final and the 0-th round $c(q_T - q_0) = c\sum_{t=1}^{T}(q_t - q_{t-1})$, Because we collect data iteratively over multiple rounds (see Figure 1), we break (1) into the sum of differences per round. Specifically in each round, we

1. Decide to grow the data set to $q_t \ge q_{t-1}$ points by sampling $\hat{\mathcal{D}} := \{\hat{z}_i\}_{i=1}^{q_t - q_{t-1}} \sim p(z)$. Pay a cost $c(q_t - q_{t-1})$ and update $\mathcal{D} \leftarrow \mathcal{D} \cup \hat{\mathcal{D}}$.
2. Train the model and evaluate the score. If $V(\mathcal{D}) \ge V^*$, then terminate.
3. If $t = T$, then pay the penalty $P$ and terminate. Otherwise, repeat for the next round.

The model score typically increases monotonically with data set size [5, 6]. This means that the minimum cost strategy for (1) is to collect just enough data such that $V(\mathcal{D}_{q_T}) = V^*$. We can estimate this minimum data requirement by modeling the score function as a stochastic process. Let $V_q := V(\mathcal{D}_q)$ and let $\{V_q\}_{q \in \mathbb{Z}_+}$ be a stochastic process whose indices represent training set sizes in different rounds. Then, collecting data in each round yields a sequence of subsampled data sets $\mathcal{D}_{q_{t-1}} \subset \mathcal{D}_{q_t}$ and their performances $V(\mathcal{D}_{q_t})$. The minimum data requirement is the stopping time

$$D^* := \arg\min_{q} \{q \mid V_q \ge V^*\}. \tag{2}$$

which is a random variable giving the first time that we pass the target. Note that $q_1^* = \cdots = q_T^* = D^*$ is a minimum cost solution to the optimal data collection problem, incurring a total cost $c(D^* - q_0)$[1].

---

[1]We assume that $c(D^* - q_0) < P$, since otherwise the optimal strategy would be to collect no data.

Estimating $D^*$ using past observations of the learning curve is difficult since we have only $T$ rounds. Further, Mahmood et al. [2] empirically show that small errors in fitting the learning curve can cause massive over- or under-collection. Thus, robust policies must capture the uncertainty of estimation.

## 4 Learn-Optimize-Collect (LOC)

Our solution approach, which we refer to as Learn-Optimize-Collect (LOC), minimizes the total collection cost while incorporating the uncertainty of estimating $D^*$. Although $D^*$ is a discrete random variable, it is realized typically on the order of thousands or greater. To simplify our problem and ensure differentiability, we assume that $D^*$ is continuous and has a well-defined density.

**Assumption 1.** *The random variable $D^*$ is absolutely continuous and has a cumulative density function (CDF) $F(q)$ and probability density function (PDF) $f(q) := dF(q)/dq$.*

In Section 4.1, we first develop an optimization model when given access to the CDF $f(q)$ and PDF $F(q)$. In Section 4.2, we estimate these distributions and combine them with the optimization model. Finally in Section 4.3, we delineate our optimization approach from prior regression methods.

### 4.1 Optimization Model

We propose an optimization problem that for any $t$, can simultaneously solve for the optimal amounts of data to collect $q_t, \ldots, q_T$ in all future rounds. Consider $t = 1$ and to develop intuition, suppose we know a priori the exact stopping time $D^*$. Then, problem (1) can be re-written as

$$\min_{q_1, \cdots q_T} \quad L(q_1, \ldots, q_T; D^*) \qquad \text{s.t.} \ \ q_0 \leq q_1 \leq \cdots \leq q_T \qquad (3)$$

where the objective function is defined recursively as follows

$$L(q_1, \ldots, q_T; D^*) := c(q_1 - q_0) + \mathbb{1}\{q_1 < D^*\}\Big(c(q_2 - q_1) + \mathbb{1}\{q_2 < D^*\}\Big(c(q_3 - q_2) \ldots$$

$$\cdots + \mathbb{1}\{q_{T-1} < D^*\}\Big(c(q_T - q_{T-1}) + P\mathbb{1}\{q_T < D^*\}\Big)\cdots\Big)\Big)$$

$$= c\sum_{t=1}^{T}(q_t - q_{t-1})\prod_{s=1}^{t-1}\mathbb{1}\{q_s < D^*\} + P\prod_{t=1}^{T}\mathbb{1}\{q_s < D^*\}$$

$$= c\sum_{t=1}^{T}(q_t - q_{t-1})\mathbb{1}\{q_{t-1} < D^*\} + P\mathbb{1}\{q_T < D^*\}.$$

The objective differs slightly from (1) due to the indicator terms, which ensure that once we collect enough data, we terminate the problem. The second line follows from gathering the terms. The third line follows from observing that $q_1 \leq q_2 \leq \cdots \leq q_T$ are constrained.

In practice, we do not know $D^*$ a priori since it is an unobserved random variable. Instead, suppose we have access to the CDF $F(q)$. Then, we take the expectation over the objective $\mathbb{E}[L(q_1, \ldots, q_T; D^*)]$ to formulate a *stochastic optimization problem* for determining how much data to collect:

$$\min_{q_1, \cdots q_T} \quad c\sum_{t=1}^{T}(q_t - q_{t-1})\left(1 - F(q_{t-1})\right) + P\left(1 - F(q_T)\right) \quad \text{s.t.} \ \ q_0 \leq q_1 \leq \cdots \leq q_T. \quad (4)$$

Note that the collection variables should be discrete $q_1, \ldots, q_T \in \mathbb{Z}_+$, but similar to the modeling of $D^*$, we relax the integrality requirement, optimize over continuous variables, and round the final solutions. Furthermore, although problem (4) is constrained, we can re-formulate it with variables $d_t := q_t - q_{t-1}$; this consequently replaces the current constraints with only non-negativity constraints $d_t \geq 0$. Finally due to Assumption 1, problem (4) can be optimized via gradient descent.

### 4.2 Learning and Optimizing the Data Requirement

Solving problem (4) requires access to the true distribution $F(q)$, which we do not have in reality. In each round, given a current training data set $\mathcal{D}_{q_t}$ of $q_t$ points, we must estimate these distribution functions $F(q)$ and $f(q)$ and then incorporate them into our optimization problem.

Given a current data set $\mathcal{D}_{q_t}$, we may sample an increasing sequence of $R$ subsets $\mathcal{D}_{q_t/R} \subset \mathcal{D}_{2q_t/R} \subset \cdots \subset \mathcal{D}_{q_t}$, fit our model to each subset, and compute the scores to obtain a data set of the learning curve $\mathcal{R} := \{(rq_t/R, V(\mathcal{D}_{rq_t/R}))\}_{r=1}^{R}$. In order to model the distribution of $D^*$, we can take $B$ bootstrap resamples of $\mathcal{R}$ to fit a series of regression functions and obtain corresponding estimates $\{\hat{D}_b\}_{b=1}^{B}$. Given a set of estimates of the data requirement, we estimate the PDF via Kernel Density Estimation (KDE). Finally to fit the CDF, we numerically integrate the PDF.

In our complete framework, LOC, we first estimate $F(q)$ and $f(q)$. We then use these models to solve problem (4). Note that in the $t$-th round of collection, we fix the prior decision variables $q_1, \ldots q_{t-1}$ constant. Finally, we collect data as determined by the optimal solution $q_t^*$ to problem (4). Full details of the learning and optimization steps, including the complete Algorithm, are in Appendix B.

### 4.3 Comparison to Mahmood et al. [2]

Our prediction model extends the previous approach of Mahmood et al. [2], who consider only point estimation of $D^*$. They (i) build the set $\mathcal{R}$, (ii) fit a parametric function $\hat{v}(q; \boldsymbol{\theta})$ to $\mathcal{R}$ via least-squares minimization, and (iii) solve for $\hat{D} = \arg\min_q \{q \mid \hat{v}(q; \boldsymbol{\theta}) \geq V^*\}$. They use several parametric functions from the neural scaling law literature, including the power law function (i.e., $\hat{v}(q; \boldsymbol{\theta}) := \theta_0 q^{\theta_1} + \theta_2$ [2, 8] where $\boldsymbol{\theta} := \{\theta_0, \theta_1, \theta_2\}$), and use an ad hoc correction factor obtained by trial and error on past tasks to help decrease the failure rate. Instead, we take bootstrap samples of $\mathcal{R}$ to fit multiple regression functions, estimate a distribution for $\hat{D}$, and incorporate them into our novel optimization model. Finally, we show in the next two sections that our optimization problem has analytic solutions and extends to multiple sources.

## 5 Analytic Solutions for the $T = 1$ Setting

In this section, we explore analytic solutions for problem (4). The unobservable $D^*$ and sequential decision-making nature suggest this problem can be formulated as a Partially Observable Markov Decision Process (POMDP) with an infinite state and action space (see Appendix C.1), but such problems rarely permit exact solution methods [32]. Nonetheless, we can derive exact solutions for the simple case of a single $T = 1$ round, re-stated below

$$\min_{q_1} \ c(q_1 - q_0) + P(1 - F(q_1)) \qquad \text{s.t. } q_0 \leq q_1 \qquad (5)$$

**Theorem 1.** *Assume $F(q)$ is strictly increasing and continuous. If there exists $q_1 \geq q_0, \hat{\epsilon} \geq 0$ where*

$$\frac{c}{P} \leq \frac{F(q_1) - F(q_0)}{q_1 - q_0}, \qquad \hat{\epsilon} \leq 1 - F(q_0), \qquad P = c/f(F^{-1}(1 - \hat{\epsilon})) \qquad (6)$$

*then there exists an $\epsilon \leq 1 - F(q_0)$ that satisfies $P = c/f(F^{-1}(1 - \epsilon))$ and an optimal solution to the corresponding problem* (5) *is $q_1^* := F^{-1}(1 - \epsilon)$. Otherwise, the optimal solution is $q_1^* := q_0$.*

When the penalty $P$ is specified via a failure risk $\epsilon$, the optimal solution to problem (5) is equal to a quantile of the distribution of $D^*$. We defer the proof and some auxiliary results to Appendix C.2.

Theorem 1 further provides guidelines on choosing values for the cost and penalty parameters. While $c$ is the dollar-value cost per-sample, which includes acquisition, cleaning, and annotation, $P$ can reflect their inherent regret or opportunity cost of failing to meet their target score. A designer can accept a risk $\epsilon$ of failing to collect enough data $\Pr\{q^* < D^*\} = \epsilon$. From Theorem 1, their optimal strategy should be to collect $F^{-1}(1 - \epsilon)$ points, which is also the optimal solution to problem (5).

## 6 The Multi-variate LOC: Collecting Data from Multiple Sources

So far, we have assumed that a designer only chooses how much data to collect and must pay a fixed per-sample collection cost. We now explore the multi-variate extension of the data collection problem where there are different types of data with different costs. For example, consider long-tail learning where samples for some rare classes are harder to obtain and thus, more expensive [33], semi-supervised learning where labeling data may cost more than collecting unlabeled data [34], or domain adaptation where a source data set is easier to obtain than a target set [35]. In this section, we highlight our main formulation and defer the complete multi-variate LOC to Appendix D.

Consider $K \in \mathbb{N}$ data sources (e.g., $K = 2$ with labeled and unlabeled) and for each $k \in \{1, \ldots, K\}$, let $z^k \sim p_k(z^k)$ be data drawn from their distribution. We train a model with a data set $\mathcal{D} := \cup_{k=1}^K \mathcal{D}^k$ where each $\mathcal{D}^k$ contains points of the $k$-th source. The performance or score function of our model is $V(\mathcal{D}^1, \ldots, \mathcal{D}^K)$. For each $k$, we initialize with $q_0^k$ points. Let $\mathbf{q}_0 = (q_0^1, \ldots, q_0^K)^\mathsf{T}$ denote the vector of data set sizes and let $\mathbf{c} = (c^1, \ldots, c^K)^\mathsf{T}$ denote costs (i.e., $c^k$ is the cost of collecting data from $p_k(z^k)$). Given a target $V^*$, penalty $P$, and $T$ rounds, we want to minimize the total cost of collection

$$\min_{\mathbf{q}_1, \ldots, \mathbf{q}_T} \mathbf{c}^\mathsf{T} \sum_{t=1}^{T} (\mathbf{q}_t - \mathbf{q}_{t-1}) + P \mathbb{1}\{V(\mathcal{D}_{q_T^1}, \ldots, \mathcal{D}_{q_T^K}) < V^*\} \quad \text{s.\,t.} \ \mathbf{q}_0 \leq \mathbf{q}_1 \leq \mathbf{q}_2 \leq \cdots \leq \mathbf{q}_T$$

We follow the same steps shown in Section 4 for this problem. First, the learning curve is now a stochastic process $\{V_\mathbf{q}\}_{\mathbf{q} \in \mathbb{Z}_+^K}$ indexed in $K$ dimensions. Next, the multi-variate analogue of the minimum data requirement in (2) is the minimum cost amount of data needed to meet the target:

$$\mathbf{D}^* := \arg\min_{\mathbf{q}} \left\{ \mathbf{c}^\mathsf{T} \mathbf{q} \mid V_\mathbf{q} \geq V^* \right\}$$

We randomly pick a unique solution to break ties. From Assumption 1, $\mathbf{D}^*$ is a random vector with a PDF $f(\mathbf{q})$ and a CDF $F(\mathbf{q}) := \int_\mathbf{0}^\mathbf{q} f(\hat{\mathbf{q}}) d\hat{\mathbf{q}}$. Finally, the multi-variate analogue of problem (4) is

$$\min_{\mathbf{q}_1, \cdots, \mathbf{q}_T} \mathbf{c}^\mathsf{T} \sum_{t=1}^{T} (\mathbf{q}_t - \mathbf{q}_{t-1}) \left(1 - F(\mathbf{q}_{t-1})\right) + P\left(1 - F(\mathbf{q}_T)\right) \ \text{s.\,t.} \ \mathbf{q}_0 \leq \mathbf{q}_1 \leq \cdots \leq \mathbf{q}_T \quad (7)$$

The Multi-variate LOC requires multi-variate PDFs, which we can fit in the same way as discussed in Section 4.2. However, we now need multi-variate regression functions that can accommodate different types of data. In Appendix D, we propose an additive family of power law regression functions that can handle an arbitrary number of $K$ sources. In our experiments, we also generalize the estimation approach of Mahmood et al. [2] to the multi-source setting for comparison.

## 7 Empirical Results

We explore the data collection problem over two sets of experiments covering single-variate $K = 1$ (Section 4) and multi-variate $K = 2$ (Section 6) problems. We consider image classification, segmentation, and object detection tasks. For every data set and task, LOC significantly reduces the number of instances where we fail to meet a data requirement $V^*$, while incurring a competitive cost with respect to the conventional baseline of naïvely estimating the data requirement [2].

In this section, we summarize the main results. We detail our data collection and experiment setup in Appendix E. We expand our full results and experiments with additional baselines in Appendix F .

### 7.1 Data and Methods

When $K = 1$, the designer decides how much data to sample without controlling the type of data. We explore classification on CIFAR-10 [36], CIFAR-100 [36], and ImageNet [37], where we train ResNets [38] to meet a target validation accuracy. We explore semantic segmentation using Deeplabv3 [39] on BDD100K [40], which is a large-scale driving data set, as well as Bird's-Eye-View (BEV) segmentation on nuScenes [41] using the 'Lift Splat' architecture [42]; for both tasks, we desire a target mean intersection-over-union (IoU). We explore 2-D object detection on PASCAL VOC [43, 44] using SSD300 [45], where we evaluate mean average precision (mAP).

When $K = 2$, the designer collects two types of data with different costs. We first divide CIFAR-100 into two subsets containing data from the first and last 50 classes, respectively. Here, we assume that the first 50 classes are more expensive to collect than the last; this mimics a real-world scenario where collecting data for some classes (e.g., long-tail) is more expensive than others. We then explore semi-supervised learning on BDD100K where the labeled subset of this data is more expensive than the unlabeled data; the cost difference between these two types is equal to the cost of data annotation.

We use a simulation model of the deep learning workflow following the procedure of Mahmood et al. [2], to approximate the true problem while simplifying the experiments (see Appendix E for full details). To avoid repeatedly sampling data, re-training a model, and evaluating the score, each

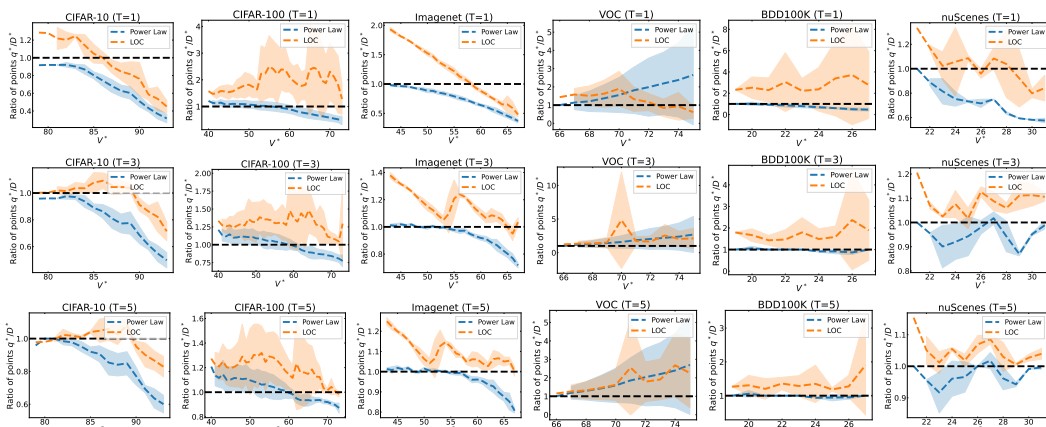

Figure 2: Mean $\pm$ standard deviation of 5 seeds of the ratio of data collected $q_T^*/D^*$ for different $V^*$. The rows correspond to $T = 1, 3, 5$ and the columns to different data sets. The black line corresponds to collecting exactly the minimum data requirement. LOC almost always remains slightly above the black line, meaning we rarely fail to meet the target.

simulation uses a piecewise-linear approximation of a 'ground truth' learning curve that returns model performance as a function of data set size. In our problems, we initialize with $q_0 = 10\%$ of the full data set (we use $20\%$ for VOC). Then in each round, we solve for the amount of data to collect and then call the piecewise-linear learning curve to obtain the current score.

We compare LOC against the conventional estimation approach of Mahmood et al. [2] who fit a regression model to the learning curve statistics, extrapolate the learning curve for larger data sets, and then solve for the minimum data requirement under this extrapolation. There are many different regression models that can be used to fit learning curves [15, 17, 5, 8]. Since power laws are the most commonly studied approach in the neural scaling law literature, we focus on these. In Appendix F.4, we show that our optimization approach can be incorporated with other regression models.

### 7.2 Main Results

We consider $T = 1, 3, 5$ rounds and $V^* \in [V(\mathcal{D}_{q_0}) + 1, V(\mathcal{D})]$ targets, where $\mathcal{D}$ is the entire data set. We evaluate all methods on (i) the failure rate, which is how often the method fails to achieve the given $V^*$ within $T$ rounds, and (ii) the cost ratio, which is the suboptimality of an algorithm for solving problem (4), i.e., $\mathbf{c}^\top(\mathbf{q}_T^* - \mathbf{q}_0)/\mathbf{c}^\top(\mathbf{D}^* - \mathbf{q}_0) - 1$. Note that the suboptimality does not count the penalty for failure since this would distort the average metrics. For $K = 1$, we also measure the ratio of points collected $q_T^*/D^*$. Although there is a natural trade-off between low cost ratio (under-collecting) and failure rate (over-collecting), we emphasize that our goal is to have low cost but with zero chance of failure.

**The Value of Optimization over Estimation when $K = 1$.** Figure 2 compares LOC versus the corresponding power law regression baseline when $c = 1$ and $P = 10^7$ ($P = 10^6$ for VOC and $P = 10^8$ for ImageNet). If a curve is below the black line, then it failed to collect enough data to meet the target. LOC consistently remains above this black line for most settings. In contrast, even with up to $T = 5$ rounds, collecting data based only on regression estimates leads to failure.

Table 1 aggregates the failure rates and cost ratios for each setting. To summarize, LOC fails at less than $10\%$ of instances for $12/18$ settings, whereas regression fails over $30\%$ for $15/18$ settings. In particular, regression nearly always under-collects data when given a single $T = 1$ round. Here, LOC reduces the risk of under-collecting by $40\%$ to $90\%$ over the baseline. While this leads to a marginal increase in costs, our cost ratios are consistently less than $0.5$ for $12/18$ settings, meaning that we spend at most $50\%$ more than the true minimum cost.

We remark that previously, Mahmood et al. [2] observed that incorrect regression estimates necessitated real machine learning workflows to collect data over multiple rounds. Instead, with LOC, we can make significantly improved data collection decisions even with a single round.

**Robustness to Cost and Penalty Parameters (see Appendix F.2 for details).** Figure 3 evaluates the ratio of points collected for $T = 5$ when the cost and the penalty of the optimization problem are

| | Data set | $T$ | Power Law Regression | | LOC | |
|---|---|---|---|---|---|---|
| | | | Failure rate | Cost ratio | Failure rate | Cost ratio |
| **Class.** | CIFAR-10 | 1 | 100% | – | **60**% | 0.19 |
| | | 3 | 95% | 0.00 | **32**% | 0.05 |
| | | 5 | 86% | 0.00 | **29**% | 0.03 |
| | CIFAR-100 | 1 | 56% | 0.12 | **4**% | 0.99 |
| | | 3 | 48% | 0.10 | **3**% | 0.31 |
| | | 5 | 48% | 0.10 | **2**% | 0.19 |
| | Imagenet | 1 | 99% | 0.00 | **37**% | 0.49 |
| | | 3 | 75% | 0.01 | **5**% | 0.16 |
| | | 5 | 56% | 0.01 | **2**% | 0.10 |
| **Seg.** | BDD100K | 1 | 77% | 0.03 | **12**% | 2.03 |
| | | 3 | 31% | 0.00 | **0**% | 0.72 |
| | | 5 | 23% | 0.01 | **0**% | 0.35 |
| | nuScenes | 1 | 95% | 0.00 | **52**% | 0.16 |
| | | 3 | 71% | 0.01 | **0**% | 0.09 |
| | | 5 | 62% | 0.00 | **0**% | 0.04 |
| **Det.** | VOC | 1 | 36% | 1.24 | **25**% | 0.56 |
| | | 3 | 8% | 0.88 | **0**% | 1.10 |
| | | 5 | 6% | 0.86 | **0**% | 0.84 |

Table 1: Average cost ratio $\mathbf{c}^\mathsf{T}(\mathbf{q}_T^* - \mathbf{q}_0)/\mathbf{c}^\mathsf{T}(\mathbf{D}^* - \mathbf{q}_0) - 1$ and failure rate measured over a range of $V^*$ for each $T$ and data set. We fix $c = 1$ and $P = 10^7$ ($P = 10^6$ for VOC and $P = 10^8$ for ImageNet). The best performing failure rate for each setting is bolded. The cost ratio is measured only for instances that achieve $V^*$. LOC consistently reduces the average failure rate, often down to 0%, while keeping the average cost ratio almost always below 1 (i.e., spending at most $2\times$ the optimal amount).

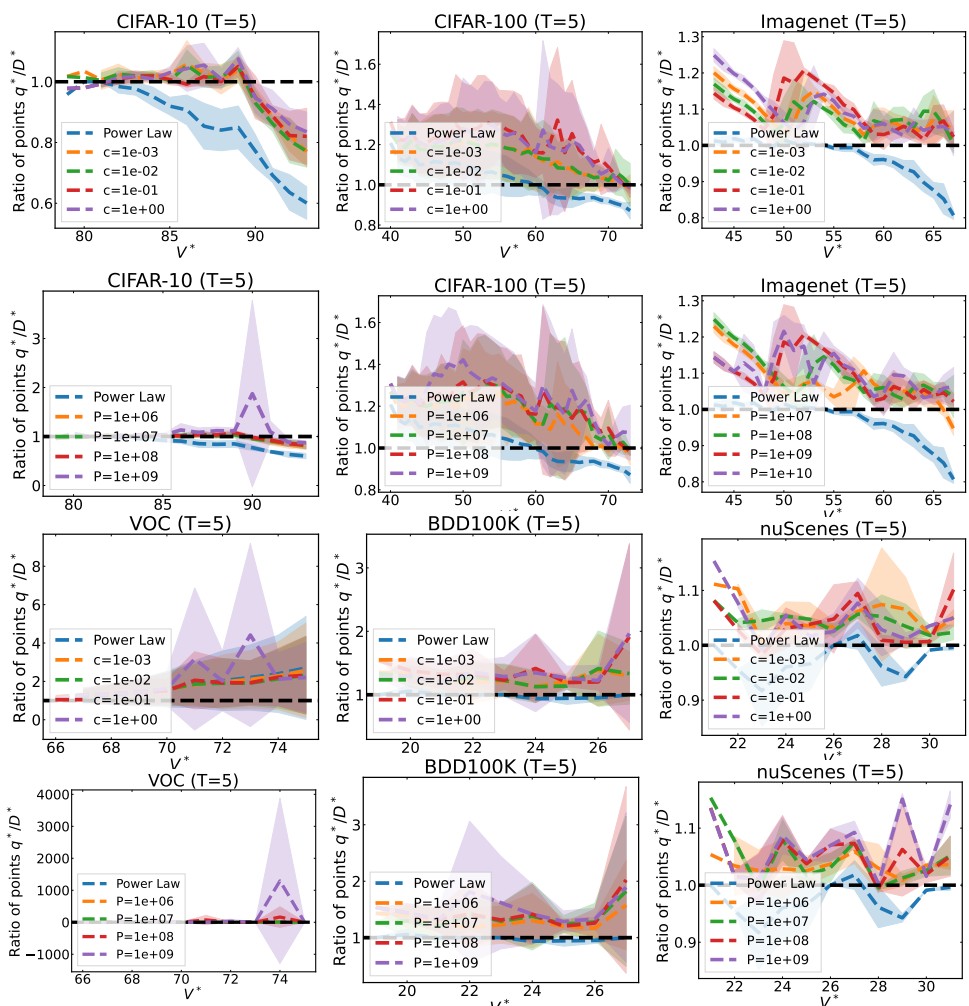

Figure 3: Mean $\pm$ standard deviation of 5 seeds of the ratio of data collected $q_T^*/D^*$ for different $V^*$ and fixed $T = 5$. *Rows 1 & 3:* We sweep the cost parameter from 0.001 to 1 and fix $P = 10^7$. *Rows 2 & 4:* We sweep the penalty parameter from $10^6$ to $10^9$ and fix $c = 1$. The dashed black line corresponds to collecting exactly the minimum data requirement. See Appendix F for all $T$.

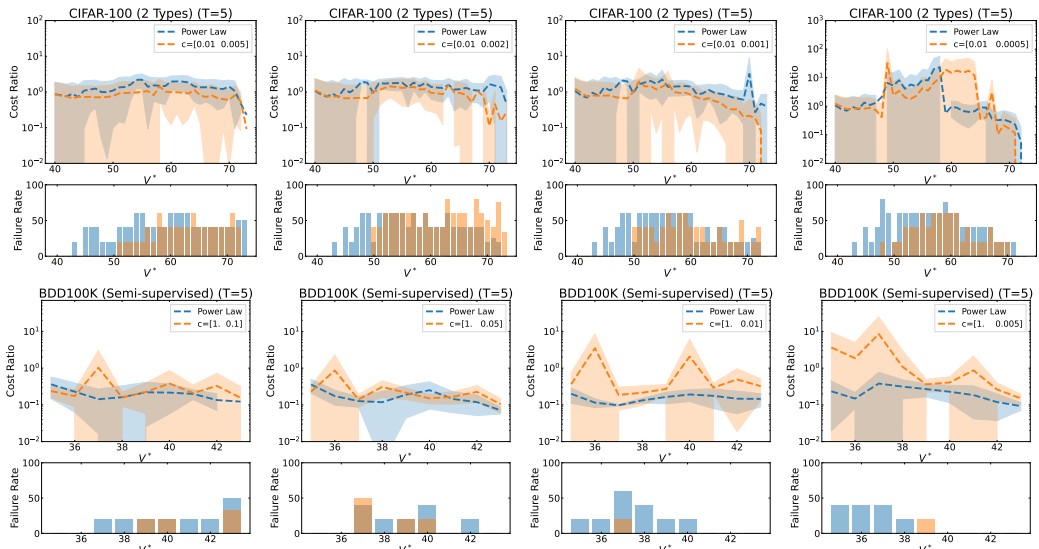

Figure 4: Mean $\pm$ standard deviation over 5 seeds of the cost ratio $\mathbf{c}^\mathsf{T}(\mathbf{q}_T^* - \mathbf{q}_0)/\mathbf{c}^\mathsf{T}(\mathbf{D}^* - \mathbf{q}_0) - 1$ and failure rate for different $V^*$, after removing 99-th percentile outliers. The columns correspond to scenarios where the first set $c^1$ costs increasingly more than the second $c^2$. See Appendix F for all $T$.

varied. Our algorithm is robust to variations in these parameters, as LOC retains the same shape and scale for almost every parameter setting and data set. Further, LOC consistently remains above the horizontal 1 line, showing that even after varying $c$ and $P$, we do not fail as frequently as the baseline. Finally, validating Theorem 1, the penalty parameter $P$ provides natural control over the amount of data collected. As we increase $P$, the ratio of data collected increases consistently.

**The Value of Optimization over Estimation when $K = 2$ (Appendix F.3).** Figure 4 compares LOC versus regression at $T = 5$ with different costs, showing that we maintain a similar cost ratio to the regression alternative, but with lower failure rates. Table 2 aggregates failure rates and cost ratios for all settings, showing LOC consistently achieves lower failure rates for nearly all settings of $T$. When $T = 5$, LOC also achieves lower cost ratios versus regression on CIFAR-100, meaning that with multiple rounds of collection, we can ensure meeting performance requirements while paying nearly the optimal amount of data. However, solving the optimization problem is generally more difficult as $K$ increases, and we sometimes over-collect data by large margins. In practice, these outliers can be identified from common sense (e.g., if a policy suggests collecting more data than we can reasonably afford, then we would not use the policy suggestion). Consequently, we report these results after removing the 99-th percentile outliers with respect to total cost for both methods. Nonetheless, this challenge remains when $T = 1$, particularly for CIFAR-100.

# 8    Discussion

We develop a rigorous framework for optimizing data collection workflows in machine learning applications, by introducing an optimal data collection problem that captures the uncertainty in estimating data requirements. We generalize this problem to more realistic settings where multiple data sources incur different collection costs. We validate our solution algorithm, LOC, on six data sets covering classification, segmentation, and detection tasks to show that we consistently meet pre-determined performance metrics regardless of costs and time horizons.

Our approach relies on estimating the CDF and PDF of the minimum data requirement, which is a challenging problem, especially with multiple data sources. Nonetheless, LOC can be deployed on top of future advances in estimating neural scaling laws. Further, we allow practitioners to input problem-specific costs and penalties, but these quantities may not always be readily available. We provide some theoretical insight into parameter selection and show that LOC is robust to these parameters. Finally, our empirical analysis focuses on computer vision, but we expect our approach to be viable in other domains governed by scaling laws.

| Data set | $T$ | Cost | Power Law Regression | | LOC | |
|---|---|---|---|---|---|---|
| | | | Failure rate | Cost ratio | Failure rate | Cost ratio |
| CIFAR-100 (2 Types) | 1 | $(0.01, 0.0005)$ | 62% | 0.89 | **40**% | 41.80 |
| | | $(0.01, 0.001)$ | 58% | 1.19 | **46**% | 9.85 |
| | | $(0.01, 0.002)$ | 56% | 1.55 | **54**% | 6.98 |
| | | $(0.01, 0.005)$ | 54% | 1.65 | **33**% | 4.43 |
| | 3 | $(0.01, 0.0005)$ | 43% | 3.47 | **30**% | 4.88 |
| | | $(0.01, 0.001)$ | 45% | 1.22 | **43**% | 1.31 |
| | | $(0.01, 0.002)$ | 45% | 1.47 | **44**% | 1.21 |
| | | $(0.01, 0.005)$ | 38% | 1.31 | **36**% | 1.17 |
| | 5 | $(0.01, 0.0005)$ | 38% | 3.31 | **24**% | 5.19 |
| | | $(0.01, 0.001)$ | 35% | 1.22 | **24**% | 0.79 |
| | | $(0.01, 0.002)$ | **37**% | 1.33 | 38% | 0.90 |
| | | $(0.01, 0.005)$ | 36% | 1.30 | **24**% | 0.82 |
| BDD100K (Semi-supervised) | 1 | $(1, 0.005)$ | 86% | 0.11 | **44**% | 7.02 |
| | | $(1, 0.01)$ | 79% | 0.15 | **30**% | 13.47 |
| | | $(1, 0.05)$ | 72% | 0.19 | **49**% | 1.02 |
| | | $(1, 0.1)$ | 70% | 0.19 | **65**% | 0.40 |
| | 3 | $(1, 0.005)$ | 23% | 0.18 | **7**% | 1.20 |
| | | $(1, 0.01)$ | 21% | 0.15 | **7**% | 2.57 |
| | | $(1, 0.05)$ | 26% | 0.18 | **23**% | 0.50 |
| | | $(1, 0.1)$ | 26% | 0.21 | **30**% | 0.15 |
| | 5 | $(1, 0.005)$ | 16% | 0.22 | **2**% | 1.91 |
| | | $(1, 0.01)$ | 21% | 0.15 | **2**% | 0.86 |
| | | $(1, 0.05)$ | 16% | 0.17 | **9**% | 0.27 |
| | | $(1, 0.1)$ | 16% | 0.20 | **7**% | 0.32 |

Table 2: Average cost ratio $\mathbf{c}^\mathsf{T}(\mathbf{q}_T^* - \mathbf{q}_0)/\mathbf{c}^\mathsf{T}(\mathbf{D}^* - \mathbf{q}_0) - 1$ and failure rate over different $V^*$ for each $T$ and $\mathbf{c}$, after removing 99-th percentile outliers. We fix $P = 10^{13}$ for CIFAR-100 and $P = 10^8$ for BDD100K. The best performing failure rate for each setting is bolded. The cost ratio is measured over instances that achieve $V^*$. LOC consistently reduces the average failure rate, and for $T > 1$, preserves the cost ratio. Further, LOC is more robust to uneven costs than regression.

Improving data collection practices yields potentially positive and negative societal impacts. LOC reduces the collection of extraneous data, which can, in turn, reduce the environmental costs of training models. On the other hand, equitable data collection should also be considered in real-world data collection practices that involve humans. We envision a potential future work to incorporate privacy and fairness constraints to prevent over- or under-sampling of protected groups. Finally, our method is guided by a score function on a held-out validation set. Biases in this set may be exacerbated when optimizing data collection to meet target performance.

There is a folklore observation that over $80\%$ of industry machine learning projects fail to reach production, often due to insufficient, noisy, or inappropriate data [46, 47]. Our experiments verify this by showing that naïvely estimating data requirements will often yield failures to meet target performances. We believe that robust data collection policies obtained via LOC can reduce failures while further guiding practitioners on how to manage both costs and time.

## Acknowledgments and Disclosure of Funding

We thank Jonathan Lorraine, Daiqing Li, Jonah Philion, David Acuna, and Zhiding Yu, as well as the anonymous reviewers and meta-reviewers for valuable feedback on earlier versions of this paper.

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
