# A  Table of Notation

| | |
|---|---|
| $T \in \mathbb{N}$ | Total number of rounds of collection. |
| $V^* \in \mathbb{R}$ | Target performance. |
| $P \in \mathbb{N}$ | Penalty for failing to reach target performance. |
| $c \in \mathbb{N}$ | Cost of per-item collection. |
| $q_t \in \mathbb{N},\ t \in \{0, \ldots, T\}$ | Number of data points in round $t$. |
| $\mathcal{D}_q$ | Data set of size $q$. |
| $V(\mathcal{D})$ | Valuation of model trained on data set $\mathcal{D}$. |
| $V_q := V(\mathcal{D}_q)$ | Short-hand notation for model valuation. |
| $D^* := \arg\min_q \{q \mid V_q \geq V^*\}$ | The minimum data requirement. |
| $F(q) = P(D^* < q)$ | The CDF of the minimum data requirement. |
| $f(q) = dF(q)/dq$ | The PDF of the minimum data requirement. |
| $L(q_1, \ldots, q_T; D^*)$ | The stochastic objective function for data collection. |
| $d_t \geq 0$ | Amount of data to collect in round $t$. |
| $\mathcal{R} := \left\{ (rq_t/R, V(\mathcal{D}_{rq_t/R})) \right\}_{r=1}^{R}$ | The regression set representing the learning curve of a given model and data set. |
| $K \in \mathbb{N}$ | Number of data sources in multi-source setting. |
| $\mathbf{q}_t \in \mathbb{N}^K$ | Vector containing number of data points per-source in round $t$. |
| $\mathbf{c} \in \mathbb{N}^K$ | Vector of per-source costs. |

Table 3: Table of notation used throughout paper.

# B  Learn-Optimize-Collect

Algorithm 1 summarizes the complete workflow of LOC within the data collection problem. Given a target score $V^*$, we collect data until we have met the target or until $T$ rounds have passed. In this section, we first expand our complete LOC algorithm. We then provide further details on the regression procedure used to estimate the data requirement distributions. Finally, we introduce a more practical reformulation of our optimization problem (4), which is what we later use in our numerical experiments.

## B.1  Details on the Learning and Optimizing Approach

Our approach for determining how much data to collect consists of three steps. The first step is to collect performance statistics by measuring the training dynamics of the current data set $\mathcal{D}_{q_t}$. Here, we sample subsets of the current data set, train our model on the subset, and evaluate the score. We repeat this process for up to $R$ subsets, where $R$ denotes the size of our performance statistics regression data set $\mathcal{R}$.

The second step is to model the data requirement distribution $D^*$. We perform this by creating bootstrap samples from $\mathcal{R}$, and then fitting a regression model $\hat{v}(q; \boldsymbol{\theta})$ that can estimate the model score as a function of data set size. We then invert this regression function by solving for the minimum $\hat{q}$ for which the model will predict that we have exceeded the target performance. We repeat this process to obtain $B$ bootstrap estimates of $D^*$. We can then fit any density estimation model to approximate a probability density and a cumulative density function. In this paper, we focus on Kernel Density Estimation (KDE) and Gaussian Mixture models since such models can be easily fit and provide functional forms of the PDF.

Finally in the third step, we solve our optimization problem (4) via gradient descent. This problem yields the optimal data set sizes $q_1^*, \ldots, q_T^*$ that we should have at the end of each round. Furthermore, if we are in the $t$-th round for $t > 1$, we freeze the values for $q_1, \ldots, q_{t-1}$ to the data set sizes that we have observed in the previous rounds. Upon solving this problem, we then collect data until we have $q_t$ samples, and then re-train our model to evaluate our current state.

---

**Algorithm 1** Learn-Optimize-Collect (LOC)

---

1: **Input:** Initial data set $\mathcal{D}_{q_0}$, Sampling distribution $p(z)$, Score function $V(\mathcal{D})$, Target score $V^*$, Maximum rounds $T$, Cost $c$, Penalty $P$, Regression model $\hat{v}(q; \boldsymbol{\theta})$, Regression set size $R$, Density Estimation model $f(q)$, Number of bootstrap samples $B$.
2: Initialize round $t = 0$, loss $L = 0$
3: **repeat**
4:     COLLECT PERFORMANCE STATISTICS
5:         Initialize $\mathcal{R} = \emptyset, \mathcal{D}_0 = \emptyset$
6:         **for** $r \in \{1, \ldots, R\}$ **do**
7:             Sub-sample $\mathcal{D}_{q_t r/R} \subset \mathcal{D}_{q_t}$ by augmenting to $\mathcal{D}_{q_t (r-1)/R}$
8:             Evaluate $V(\mathcal{D}_{q_t r/R})$ and update $\mathcal{R} \leftarrow \mathcal{R} \cup \{(q_t r/R, V(\mathcal{D}_{q_t r/R})\}$
9:         **end for**
10:    **end**
11:    ESTIMATE DATA REQUIREMENT DISTRIBUTION
12:         Initialize $\mathcal{Q} = \emptyset$
13:         **for** $b \in \{1, \ldots, B\}$ **do**
14:             Create bootstrap $\mathcal{R}_b$ by sub-sampling with replacement from $\mathcal{R}$
15:             Fit regression model $\boldsymbol{\theta}^* = \arg\min_{\boldsymbol{\theta}} \sum_{(q,v) \in \mathcal{R}_b} (v - \hat{v}(q; \boldsymbol{\theta}))^2$
16:             Estimate data requirement $\hat{q}_b = \arg\min_q \{q \mid \hat{v}(q; \boldsymbol{\theta}^*) \geq V^*\}$
17:             Update $\mathcal{Q} \leftarrow \mathcal{Q} \cup \{\hat{q}_b\}$
18:         **end for**
19:         Fit Density Estimation model $f(q)$ using the empirical distribution $\mathcal{Q}$ and let $F(q) = \int_0^q f(q)dq$
20:    **end**
21:    COLLECT DATA
22:         Solve problem (4) using Density Estimation models $f(q)$ and $F(q)$ to obtain $q_1^*, \ldots, q_T^*$.
23:         Sample $\hat{\mathcal{D}} \sim p(z)$ until $|\hat{\mathcal{D}}| = q_t - q_{t-1}$
24:         Update loss $L \leftarrow L + c(q_t - q_{t-1})$
25:         Update $\mathcal{D}_{q_t} = \mathcal{D}_{q_{t-1}} \cup \hat{\mathcal{D}}$
26:    **end**
27:    Evaluate $V(\mathcal{D}_{q_t})$ and update $t \leftarrow t + 1$
28: **until** $V(\mathcal{D}_{q_t}) \geq V^*$ or $t = T$
29: **if** $V(\mathcal{D}_{q_t}) < V^*$ **then**
30:    Update loss $L \leftarrow L + P$
31: **end if**
32: **Output:** Final collected data set $\mathcal{D}_{q_t}$, loss $L$

---

## B.2 Regression Model for $D^*$

To estimate the data requirement, we build a regression model of the learning curve and then invert this curve. The classical learning curve literature proposes using structured functions to regress on the learning curve. For example with neural scaling laws, the most common function is a power law model $\hat{v}(q; \boldsymbol{\theta}) := \theta_0 q^{\theta_1} + \theta_2$ where $\boldsymbol{\theta} := \{\theta_0, \theta_1, \theta_2\}$ [5, 18, 16, 6, 9, 10, 19, 8, 2]. As a result, the main experiments of our paper use this model.

We fit each regression function by minimizing a least squares problem using the Levenberg-Marquardt algorithm as implemented by Scipy [48, 49]. The parameters for each function are initialized to either 1 or 0 depending on if they are product or bias terms. To further help fit the data, we use weighted least squares where each subsequent point is weighted twice as much as the previous point. This ensures that our regression model is tuned to better fit the curve for larger $q$.

## B.3 Alternate Re-formulation of Problem (4)

Although problem (4) has a differentiable objective function, it includes a set of lower bound constraints, meaning that a vanilla gradient descent algorithm may not immediately apply. A naive solution algorithm may be to use projected gradient descent. Instead, we show that the problem can be reformulated to remove these constraints.

For each $t$, let $d_t = q_t - q_{t-1}$ denote the additional amount of data collected in each round. Note that we can recursively re-define $q_t = q_0 + \sum_{r=1}^t d_r$. Furthermore, the ordering inequality constraints can all be re-written to non-negativity constraints $d_t \geq 0$. We now re-write problem (4) using these

new variables:

$$\min_{d_1, \cdots d_T} \quad c \sum_{t=1}^{T} d_t \left( 1 - F \left( q_0 + \sum_{r=1}^{t-1} d_r \right) \right) + P \left( 1 - F \left( q_0 + \sum_{r=1}^{T} d_r \right) \right)$$
$$\text{s.t.} \quad d_1, \ldots, d_T \geq 0.$$

The objective in the above formulation remains differentiable. Moreover, the non-negativity constraints can be removed by re-writing $d_t \leftarrow \mathrm{softplus}(d_t)$ for all $t$. As a result, we can solve this problem by using an off-the-shelf gradient descent algorithm. Finally we note that in our experiments, we implemented a projected gradient descent based on (4) and the above gradient descent algorithm. We found that the above streamlined approach to generate slightly better final solutions.

## C  Details on the Main Theory

In this section, we explore mathematical properties and challenges of the optimal data collection problem. In Section C.1, we discuss the drawbacks of alternative approaches to our modeling framework. Then in Section C.2, we prove our main theorem demonstrating an analytic solution in the single-round problem.

### C.1  Markov Decision Process Alternatives

The data collection problem requires sequential decision-making in terms of collecting additional data in each round. A natural modeling approach for such problems is via Markov Decision Processes (MDPs). However, MDP techniques are challenging for this problem due to (i) the unobservability of the state, and (ii) an infinite state space. That is, until we have collected enough data to meet the target, we do not know how much data is the minimum, which can furthermore be arbitrarily large. Here, we sketch a potential alternative MDP framework and highlight the core challenge.

Our sequential decision-making problem can be written as a Partially Observable Markov Decision Process (POMDP) [50, 51]. Furthermore, because this state variable is constant throughout the collection problem, we can write it as an EK 'Learning-and-Doing' model [52]. Such POMDPs are defined by the tuple $(\Theta, \mathcal{A}, \mathcal{S}, p, r_t)$, where the state space characterizes the data requirement $D^* \in \Theta := \mathbb{R}_+$, the action space characterizes the additional data collected $d_t := (q_t - q_{t-1}) \in \mathcal{A} := \mathbb{R}_+$, and the observation set $\mathcal{S} := \{0, 1\}$ characterizes a binary variable $\mathbb{1}\{V(\mathcal{D}_{q_t}) \geq V^*\} = \mathbb{1}\{q_t \geq D^*\}$. Furthermore, $p(\cdot|D^*, d_t)$ is the observation transition probability and $r_t(\cdot)$ is the reward function where $r_t(q_t, q_{t-1}, D^*) := -c(q_t - q_{t-1})$ for $t \leq T$ and $r_{T+1}(q_t, q_{t-1}, D^*) := -P\mathbb{1}\{q_T < D^*\}$.

Because the state variable is unobserved, POMDPs are typically solved by using a belief distribution of the state variable to average the reward in the value function. When both the state and the action space of a POMDP are finite, Smallwood and Sondik [53] show that this value function is piecewise-linear and can be solved by exact methods, albeit under a curse of dimensionality with respect to these spaces. Unfortunately, for a general POMDP with an infinite state and action space, these methods do not apply and we most often resort to approximation techniques [54]. Moreover in our case, approximations based on discretizing the state and action space fall prey to the curse of dimensionality.

Alternatively, we may naively consider applying reinforcement learning. However, note that real-world data collection tasks do not contain the requisite sizes of learning data or generalizable simulation mechanisms that are staples in reinforcement learning techniques. These challenges, coupled with the goal of delivering practical managerial guidelines for data collection operations, motivate us to explore easy-to-implement techniques for optimizing data collection.

### C.2  Proof of Theorem 1

Our main theorem states that the one-round problem has an analytic solution. However, the proof requires several auxiliary results. For clarity, we first reproduce the theorem.

**Theorem 1 (Repeated).** *Consider the one-round problem*

$$\min_{q_1} \ c(q_1 - q_0) + P(1 - F(q_1)) \qquad \text{s.t.} \ q_0 \leq q_1$$

*Assume $F(q)$ is strictly increasing and continuous. For any $\epsilon$ such that $F(q_0) < 1 - \epsilon$, let $P :=$ $c/f(F^{-1}(1 - \epsilon))$. The optimal solution to the corresponding problem (5) is $q_1^* = F^{-1}(1 - \epsilon)$. Furthermore, the optimal solution satisfies $F(q_1^*) = 1 - \epsilon$.*

The assumption of a strictly increasing and continuous cumulative density function is needed to ensure that the data requirement distribution has a well-defined quantile function $F^{-1}(p) :=$ $\inf_q\{q \mid F(q) \geq p\}$, where the optimal solution for any $p \in (0, 1)$ is unique.

The proof for Theorem 1 relies on equating the one-round optimization problem to the following constrained optimization problem:

$$\min_{q_1} \ c(q_1 - q_0) \qquad \text{s.t. } F(q_1) \geq 1 - \epsilon, \ \ q_0 \leq q_1 \tag{8}$$

where $\epsilon > 0$ is a pre-determined parameter. This above problem (8) is solving for the least amount of additional data to collect such that with probability at least $1 - \epsilon$, we collect above the minimum data requirement. We first characterize the properties of problem (8).

**Lemma 1.** *Problem* (8) *is a convex optimization problem.*

*Proof.* We only need to prove that the set $\{q_1 \mid F(q_1) \geq 1 - \epsilon\}$ is a convex set, since the objective and remaining constraint are convex. Since $F(q)$ is a monotonically non-decreasing function in $q$, for any $\theta \in [0, 1]$ and $q < \hat{q}$ that satisfy the CDF constraint, we have

$$F(\theta q + (1 - \theta)\hat{q}) \geq F(q) \geq 1 - \epsilon.$$

Because the convex combination of any two points is in the set, the set must be convex. $\qquad \square$

**Lemma 2.** *The optimal solution to problem* (8) *is*

$$q^* = \begin{cases} F^{-1}(1 - \epsilon), & \text{if } F(q) < 1 - \epsilon \\ q_0 & \text{otherwise} \end{cases}$$

*Proof.* First consider the case where $F(q_0) \geq 1 - \epsilon$. Then, $q_0$ is a feasible solution to problem (8). Furthermore due to the second constraint, any $q < q_0$ is infeasible. Since $q_0$ minimizes the objective, it is optimal.

Next consider the case where $F(q_0) < 1 - \epsilon$. Then, let $q_1 = F^{-1}(1 - \epsilon) = \inf\{q \mid F(q) \geq 1 - \epsilon\}$ be the smallest solution that satisfies the CDF constraint. By the monotonicity of $F(q)$, it follows that $q_1 > q_0$. Therefore, this solution minimizes the objective. $\qquad \square$

We are now ready to prove Theorem 1.

*Proof of Theorem 1.* We prove this result by first developing two different characterizations of the optimal solution set of problem (5) and then applying their equivalence.

First note that problem (5) is an optimization problem with one variable $q_1$ over a constrained domain $[q_0, \infty)$. Furthermore, the objective function is continuous everywhere, meaning that there are two possible scenarios:

- $q_1^*$ satisfies $f(q_1) = c/P$.

- $q_1^* := q_0$ and the optimal value is $P(1 - F(q_0))$.

The first scenario is obtained by taking the derivative of the objective function and setting it to $0$. The second scenario comes from the boundary condition. Moreover, note that the objective is unbounded as $q_1 \to +\infty$ meaning the above scenario is the only boundary condition that we need to consider.

Next, note that problem (5) is equivalent to the following optimization problem

$$\begin{aligned} \min_{q_1, \epsilon} \quad & c(q_1 - q_0) + P\epsilon \\ \text{s.t.} \quad & \epsilon \geq 1 - F(q_1) \\ & q_1 \geq q_0 \end{aligned} \tag{9}$$

$$\min_{\epsilon} c(F^{-1}(1-\epsilon) - q_0) + P\epsilon \qquad \text{s.\,t.\,} \epsilon \geq 1 - F(q_0)$$

For any fixed $\epsilon$, problem (9) is equivalent to problem (8), and therefore by Lemma 1, the problem is convex optimization problem.

We can optimize problem (9) by breaking into two cases. First, for any fixed $\epsilon \geq 1 - F(q_0)$, setting $q_1^*(\epsilon) = 0$ attains a feasible solution and mnimizes the objective to $P\epsilon$.

Second, for any fixed $\epsilon \leq 1 - F(q_0)$, Lemma 2 states that $q_1^*(\epsilon) = F^{-1}(1-\epsilon)$ is a corresponding optimal solution and the objective function reduces to

$$c\left(F^{-1}(1-\epsilon) - q_0\right) + P\epsilon.$$

Moreover, from the original formulation (5), we can substitute $q_1^*(\epsilon)$ and obtain $f(F^{-1}(1-\epsilon) = c/P$.

Finally, problem (9) is optimized via the second case if and only if there exists a feasible $\epsilon \leq 1 - F(q_0)$ that satisfies

$$c\left(F^{-1}(1-\epsilon) - q_0\right) + P\epsilon \leq P(1 - F(q_0)).$$

We can rewrite this condition as follows. Let $q_1 > q_0$ and assume that (6) holds. Then,

$$
\begin{aligned}
& c(q_1 - q_0) \leq PF(q_1) - PF(q_0) \\
\Rightarrow \quad & c(q_1 - q_0) - PF(q_1) \leq -PF(q_0) \\
\Rightarrow \quad & c(q_1 - q_0) + P(1 - F(q_1)) \leq P(1 - F(q_0)).
\end{aligned}
$$

Let $\epsilon = 1 - F(q_1)$. Since $F(q_1) \geq F(q_0)$, we have $\epsilon \leq 1 - F(q_0)$, meaning that there is a feasible $q_1 > q_0$ to problem (9) with lower objective function value than $q_0$. Thus, assumption (6) guarantees that problem (9) has an optimal solution $q_1^* = F^{-1}(1-\epsilon^*)$ where $\epsilon^*$ must satisfy $f(F^{-1}(1-\epsilon^*) = c/P$. Conversely, if (6) is not satisfiable for any $q_1 > q_0$, then we can use the same steps to show that $q_1^* = q_0$ is an optimal solution to the problem. $\qquad \square$

## D  Optimal Data Collection with Multiple Sources

The multi-variate data collection problem considers multiple sources delivering different types of data required to train a model. Consider $K$ data sets with $q^1, \ldots, q^K$ points in each, respectively. Rather than collecting up to $q_t$ data points in each round, we optimize a vector $\mathbf{q}_t \in \mathbb{R}_+^K$ where each element $q_t^k$ refers to how much data we need from the $k$-th source. Furthermore, the minimum data requirement is now a vector $\mathbf{D}^*$.

This problem can be solved using the same general approach outlined in Algorithm 1, but with two changes. First, we require a multi-variate version of the PDF and CDF of $\mathbf{D}^*$. This necessitates new neural scaling law regression models for dealing with multiple data sources. Second, we modify the optimization problem from Appendix B to accommodate decision vectors. We highlight the above two steps in this section.

### D.1  A General Multi-variate Neural Scaling Law

In order to construct a PDF and CDF in the multi-variate setting, we follow the same general steps as in Algorithm 1. We first collect a data set of performance statistics $\mathcal{R} := \{(\mathbf{q}_r, V(\mathcal{D}_{q_r^1}^1, \mathcal{D}_{q_r^2}^2, \cdots, \mathcal{D}_{q_r^K}^K))\}_{r=1}^R$ as before. We then use bootstrap resamples of this data set to fit parameters $\boldsymbol{\theta}^*$ to a regression model $\hat{v}(q^1, \ldots, q^K; \boldsymbol{\theta})$ and then solve for

$$\hat{\mathbf{q}} := \arg\min_{\mathbf{q}} \{\mathbf{c}^\mathsf{T}\mathbf{q} \mid \hat{v}(q^1, \ldots, q^K; \boldsymbol{\theta}^*) \geq V^*\}.$$

Finally, we fit a density estimation model over our data set of $\hat{\mathbf{q}}$.

The key challenge to this approach however is in designing a multi-variate regression function. To the best of our knowledge, the neural scaling law literature has not explored general power law models that can accommodate $K$ different types of data for arbitrary tasks.

We propose an easy-to-implement baseline regression model by adding the contributions of each data set being used. Then, our additive regression model is

$$\hat{v}(q^1, \ldots, q^K; \boldsymbol{\theta}) := \sum_{k=1}^{K} \hat{v}_k(q^k; \boldsymbol{\theta}_k)$$

where $\hat{v}_k(q_k; \boldsymbol{\theta}_k)$ can be any single-variate regression model for estimating score. For instance, consider $K = 2$ data types with power law regression models for each data type. Our multi-variate regression model becomes

$$\hat{v}(q^1, q^2; \boldsymbol{\theta}) = \theta_{1,0}(q^1)^{\theta_{1,1}} + \theta_{2,0}(q^2)^{\theta_{2,1}} + \theta_3.$$

We may also incorporate other base models in the same way, such as the logarithmic or arctan functions introduced in Mahmood et al. [2].

The benefit of this additive regression model is that we can easily fit it via least squares minimization using a regression data set $\mathcal{R} := \{(q_r^1, q_r^2, V(\mathcal{D}_{q_r^1}^1, \mathcal{D}_{q_r^2}^2)\}_{r=1}^{R}$. Furthermore, additive models are simple and offer interpretable explanations on the contributions of each data type to performance by assuming that each data set has an independent effect. Finally, additive models are common in many other tasks, such as when estimating the valuation of specific data points [55].

We remark that some recent research has explored neural scaling laws for specific tasks with multiple data types. For instance, Mikami et al. [11] explore a $K = 2$ power law for transfer learning from synthetic to real domains, where they use a multiplicative component that captures an interaction between real and synthetic data sets. Because scaling laws in multi-variate settings remain an open area of study, if there exist specific structural regression functions for a given application with different types of data, then such functions should be used in place of the additive model. Moreover, our downstream optimization model operates independently of the regression model, as long as the regression model can be re-trained with bootstrap samples in order to facilitate density estimation.

### D.2 The Optimization Problem with Multiple Decisions

Just as in the single-variate case, problem (7) has a differential objective function but a series of lower bound constraints. We can use the same approach highlighted in Appendix B to reformulate this problem and remove the constraints. We summarize this reformulation below.

For each $t$, let $\mathbf{d}_t = \mathbf{q}_t - \mathbf{q}_{t-1}$ be the additional data collected in each round. Then, we recursively re-define $\mathbf{q}_t = \mathbf{q}_0 + \sum_{r=1}^{t} \mathbf{d}_r$ and re-write the problem to

$$\min_{\mathbf{d}_1, \cdots \mathbf{d}_T} \quad \mathbf{c}^{\mathsf{T}} \sum_{t=1}^{T} \mathbf{d}_t \left( 1 - F \left( \mathbf{q}_0 + \sum_{r=1}^{t-1} \mathbf{d}_r \right) \right) + P \left( 1 - F \left( \mathbf{q}_0 + \sum_{r=1}^{T} \mathbf{d}_r \right) \right)$$
$$\text{s.t.} \quad \mathbf{d}_1, \ldots, \mathbf{d}_T \geq 0.$$

The above problem can now be solved using off-the-shelf gradient descent.

## E Simulation Experiment Setup

The most intuitive approach of validating our data collection problem is by repeatedly sampling from a data set, training a model, and solving the optimization problem. However, performing a large set of such experiments over many data sets becomes computationally intractable. Instead, we follow the approach introduced in Mahmood et al. [2], which proposes a simulation model of the data collection problem. This section summarizes the simulation setup.

The simulation replicates the steps in Algorithm 1 except with one key difference. In the simulation, we replace the score function $V(\mathcal{D})$ with a *ground truth* function $v_{\text{gt}}(q)$ that serves as an oracle which reports the expected score of the model trained with $q$ data points. Thus, rather than having to collect data and train a model in each round, we evaluate $v_{\text{gt}}(q_t)$ and treat this as the current model score. The optimization and regression models do not have access to $v_{\text{gt}}(q)$.

### E.1 A Piecewise-Linear Ground Truth Approximation

In order to build a ground truth function, we first use the sub-sampling procedure in Algorithm 1 to collect performance statistics over subsets of the entire training data set. Using these observed statistics, we then build a piecewise-linear model of the ground truth. Below, we first highlight how to construct a piecewise-linear model when given a set of data set sizes and their corresponding scores. In the next subsection, we will detail the exact data collection process.

**The Single-variate ($K = 1$) Case.** Mahmood et al. [2] develop a ground truth function as follows. Let $q_0 \leq q_1 \leq q_2 \leq \cdots$ be a series of data set sizes and let $\mathcal{D}_{q_0} \subset \mathcal{D}_{q_1} \subset \mathcal{D}_{q_2} \subset \cdots$ be their corresponding sets. Then, consider the following piecewise-linear function:

$$
v_{\text{gt}}(q) := \begin{cases} \dfrac{V(\mathcal{D}_{q_0})}{q_0} n, & q \leq q_0 \\ \dfrac{V(\mathcal{D}_{q_t}) - V(\mathcal{D}_{q_{t-1}})}{q_t - q_{t-1}} (q - q_t) + V(\mathcal{D}_{q_{t-1}}), & q_{t-1} \leq q \leq q_t \end{cases}
$$

This function is concave and monotonically increasing, which follows the general trend of real learning curves [5]. Furthermore Mahmood et al. [2] show that given sufficient resolution, i.e., enough data subsets, this piecewise linear function is an accurate approximation of the true learning curve $V(\mathcal{D})$.

**The Multi-variate ($K = 2$) Case.** In the previous $K = 1$ case, the ground truth was formed by taking linear approximations between different subset sizes. When $K > 1$, we have multiple subsets that are used to evaluate the score $V(\mathcal{D}^1, \ldots, \mathcal{D}^K)$.

We focus specifically on $K = 2$ in our numerical experiments and propose a generalization of the previous piecewise-linear function. Here, rather than building lines on the intervals between subsequent sets, we build planes on triangular intervals. Specifically, let $q_0^1 \leq q_1^1 \leq q_2^1 \leq \cdots$ and $q_0^2 \leq q_1^2 \leq q_2^2 \leq \cdots$ be two series of data set sizes, and consider the grid

$$
\begin{array}{cccc} (q_0^1, q_0^2) & (q_1^1, q_0^2) & (q_2^1, q_0^2) & \cdots \\ (q_0^1, q_1^2) & (q_1^1, q_1^2) & (q_2^1, q_1^2) & \cdots \\ (q_0^1, q_2^2) & (q_1^1, q_2^2) & (q_2^1, q_2^2) & \cdots \\ \vdots & \vdots & \vdots & \ddots \end{array}
$$

For each tuple $(q_s^1, q_t^2)$ in the above grid, let $V(\mathcal{D}_{q_s^1}^1, \mathcal{D}_{q_t^2}^2)$ be the score of a model trained on two data sets of the corresponding respective sizes.

For each index $(s, t)$ we fit linear models on the corresponding lower right and upper left triangles. First, let $(\underline{\alpha}(s,t), \underline{\beta}(s,t), \underline{\gamma}(s,t))$ be the parameters of the plane defined by the lower triangle $\{(q_{s-1}^1, q_t^2), (q_s^1, q_{t-1}^2), (q_s^1, q_t^2)\}$, i.e., the unique solution to the following linear system:

$$
\begin{pmatrix} q_{s-1}^1 & q_t^2 & 1 \\ q_s^1 & q_{t-1}^2 & 1 \\ q_s^1 & q_t^2 & 1 \end{pmatrix} \begin{pmatrix} \underline{\alpha}(s,t) \\ \underline{\beta}(s,t) \\ \underline{\gamma}(s,t) \end{pmatrix} = \begin{pmatrix} V(\mathcal{D}_{q_{s-1}^1}^1, \mathcal{D}_{q_t^2}^2) \\ V(\mathcal{D}_{q_s^1}^1, \mathcal{D}_{q_{t-1}^2}^2) \\ V(\mathcal{D}_{q_s^1}^1, \mathcal{D}_{q_t^2}^2) \end{pmatrix}
$$

Thus, for any data set sizes $(q^1, q^2)$ in this triangle, we evaluate the ground truth by the linear model $\underline{\alpha}(s,t)q^1 + \underline{\beta}(s,t)q^2 + \underline{\gamma}(s,t)$. Similarly, let $(\overline{\alpha}(s,t), \overline{\beta}(s,t), \overline{\gamma}(s,t))$ be the parameters of the plane defined by the upper triangle $\{(q_s^1, q_t^2), (q_{s+1}^1, q_t^2), (q_s^1, q_{t+1}^2)\}$, i.e., the unique solution to the following linear system:

$$
\begin{pmatrix} q_s^1 & q_t^2 & 1 \\ q_{s+1}^1 & q_t^2 & 1 \\ q_s^1 & q_{t+1}^2 & 1 \end{pmatrix} \begin{pmatrix} \overline{\alpha}(s,t) \\ \overline{\beta}(s,t) \\ \overline{\gamma}(s,t) \end{pmatrix} = \begin{pmatrix} V(\mathcal{D}_{q_s^1}^1, \mathcal{D}_{q_t^2}^2) \\ V(\mathcal{D}_{q_{s+1}^1}^1, \mathcal{D}_{q_t^2}^2) \\ V(\mathcal{D}_{q_s^1}^1, \mathcal{D}_{q_{t+1}^2}^2) \end{pmatrix}
$$

Similarly for any $(q^1, q^2)$ in this triangle, the ground truth is obtained by the linear model $\overline{\alpha}(s,t)q^1 + \overline{\beta}(s,t)q^2 + \overline{\gamma}(s,t)$.

Finally, we define our ground truth function $v_{\text{gt}}(q^1, q^2)$. For any $q^1 \geq q_0^1, q^2 \geq q_0^2$, this function first identifies the interval $[q_s^1, q_{s+1}^1] \times [q_t^2, q_{t+1}^2]$ in which the point lies. Then, the function assigns a

| Data set | Task | Score | Full data set size | |
|---|---|---|---|---|
| CIFAR-10 [36] | Classification | Accuracy | 50,000 | |
| CIFAR-100 [36] | Classification | Accuracy | 50,000 | |
| ImageNet [37] | Classification | Accuracy | 1,281,167 | |
| BDD100K [40] | Semantic Segmentation | Mean IoU | 7,000 | |
| nuScenes [41] | BEV Segmentation | Mean IoU | 28,130 | |
| VOC [43, 44] | 2-D Object Detection | Mean AP | 16,551 | |
| CIFAR-100 [36] | Classification | Accuracy | 25,000 (Classes 0-49) | 25,000 (Classes 50-99) |
| BDD100K [40] | Semantic Segmentation | Mean IoU | 7,000 (Labeled) | 70,000 (Unlabeled) |

Table 4: Data sets, tasks, and score functions considered.

score based on whether the point lies in the upper left or the lower right triangle in this interval. We write this function as

$$
v_{\mathrm{gt}}(q^1, q^2) := \begin{cases} \overline{\alpha}(s,t)q^1 + \overline{\beta}(s,t)q^2 + \overline{\gamma}(s,t) \\ \qquad \text{if } \left\| (q^1, q^2) - (q_s^1, q_t^2) \right\| \leq \left\| (q^1, q^2) - (q_{s+1}^1, q_{t+1}^2) \right\|, \\ \underline{\alpha}(s+1, t+1)q^1 + \underline{\beta}(s+1, t+1)q^2 + \underline{\gamma}(s+1, t+1) \\ \qquad \text{otherwise}, \end{cases}
$$
$$
\text{for } q_s^1 \leq q^1 \leq q_{s+1}^1 \ , \ q_t^2 \leq q^2 \leq q_{t+1}^2.
$$

**For $K > 2$.** The piecewise linear approximations grow increasingly complex as the dimension $K$ increases. Furthermore, the number of subsets of data set sizes required to create a piecewise linear approximation increases exponentially with $K$. Specifically for $k \in \{1, \ldots, K\}$, let $M_k$ denote the number of subsets (i.e., $|\{q_0^k, q_1^k, \ldots, q_{M_k}^k\}|$) of a data set that we consider when creating subsets. For each combination of $K$ subsets, we must then train a model and evaluate it's performance to record $V(\mathcal{D}^1, \ldots, \mathcal{D}^K)$. Thus, we must subsample and train our model for $O(\prod_k M_k)$ combinations. This can quickly become computationally prohibitive.

### E.2 Data Collection

We now summarize the data collection and training process used to create the above piecewise-linear functions for each data set and task. All models were implemented using PyTorch and trained on machines with up to eight NVIDIA V100 GPU cards. Table 4 details each task and data set size.

**Image Classification Tasks.** For all experiments with CIFAR-10 and CIFAR-100, we use a ResNet18 [38] following the same procedure as in Coleman et al. [56]. For ImageNet, we use a ResNet34 [38] using the procedure in Coleman et al. [56]. All models are trained with cross entropy loss using SGD with momentum. We evaluate all models on Top-1 Accuracy.

For all experiments, we set the initial data set at $q_0 = 10\%$ of the data. In data collection, we create five subsets containing $2\%, 4\%, \cdots, 10\%$ of the training data, five subsets containing $12\%, 14\%, \cdots, 20\%$ of the training data, and eight subsets containing $30\%, 40\%, \cdots, 100\%$ of the data. Note that we use higher granularity in the early stage as this is where the dynamics of the learning curve vary the most. With more data, the learning curve eventually has a nearly zero slope. For each subset, we train our respective model and evaluate performance.

**VOC.** We use the Single-Shot Detector 300 (SSD300) [45] based on a VGG16 backbone [57], following the same procedure as in Elezi et al. [58]. All models are trained using SGD with momentum. We evaluate all models on mean AP.

For all experiments, we set the initial data set at $q_0 = 10\%$ of the data. In data collection, we sample twenty subsets at $5\%$ intervals, i.e., $5\%, 10\%, 15\%, \cdots, 100\%$ of the training data.

**BDD100K.** We use Deeplabv3 [39] with ResNet50 backbone. We use random initialization for the backbone. We use the original data set split from Yu et al. [40] with $7,000$ and $1,000$ data points in the train and validation sets respectively. The evaluation metrics is mean Intersection over Union (IoU). We follow the same protocol used in the Image classification tasks to create our subsets of data.

**nuScenes.** We use the "Lift Splat" architecture [42], which is used for BEV segmentation from driving scenes, following the steps from the original paper to train this model. We evaluate on mean

| Parameter | Setting |
|---|---|
| Optimizer | GD with Momentum ($\beta = 0.9$), Adam ($\beta_0, \beta_1 = 0.9, 0.999$) |
| Learning rate | $0.005, \ldots, 500$ |
| Number of bootstrap samples $B$ | 500 |
| Number of regression subsets $R$ | See Appendix E.2 |
| Density Estimation Model | KDE for $K = 1$, GMM for $K = 2$ |
| KDE Bandwidth | $20000, \ldots, 20000000$ for ImageNet $200, \ldots, 4000$ for all others |
| GMM number of clusters | $4, \ldots, 10$ |

Table 5: Summary of hyperparameters used in our experiments.

IoU. Our data collection procedure follows the same steps and percentages of the data set as used for BDD100K and the Image classification tasks.

**CIFAR-100 (2 Types).** We partition this data set into two subsets $\mathcal{D}^1$ and $\mathcal{D}^2$ of $25,000$ images each containing the first 50 and last 50 classes, respectively. We then train a ResNet18 [38] using different fractions of the two subsets. We follow the same training procedure as in the single-variate case except with one difference. Since some of the data sets will naturally be imbalanced (e.g., if we train with half of the first subset and all of the second subset), we employ a class-balanced cross entropy loss using the inverse frequency of samples per class.

For each $\mathcal{D}^k$ subsets, respectively, we follow the same subsampling procedure used in the single-variate case. That is, we let $q_0^1 = 10\%$ of the first data subset and $q_0^2 = 10\%$ of the second data subset. For each subset, we create 10 subsampled sets at intervals of $2\%, 4\%, 6\%, \cdots, 20\%$ of the respective data subset. We then create eight further subsampled sets at $30\%, 40\%, \cdots, 100\%$ of the respective data subset. Finally, we train our model and evaluate the score on every combination of the subsampled subsets of $\mathcal{D}^1 \times \mathcal{D}^2$.

**BDD100K (Semi-supervised).** For this task, we consider semi-supervised segmentation via pseudo-labeling the unlabeled data set in BDD100K. The data is partitioned into two subsets $\mathcal{D}^1$ and $\mathcal{D}^2$ containing $7,000$ labeled and $70,000$ unlabeled scenes. As before, we use the Deeplabv3 [39] architecture with a ResNet50 backbone. Here however, we:

1. First train with a labeled subset of $\mathcal{D}^1$ via supervised learning.

2. Pseudo-label an unlabeled subset of $\mathcal{D}^2$ using the trained model.

3. Re-train the model with the labeled subset and the pseudo-labeled subset.

We follow the same procedure as in the single-variate case for both training steps, except we weigh the unlabeled data by $0.2$ to reduce its contribution to the loss.

Training via semi-supervised learning on BDD100K requires long compute times, so we reduce the number of subsets used in this experiment. For the labeled set $\mathcal{D}^1$, we create subsets with $5\%, 10\%, 15\%, 20\%, 40\%, 60\%, 80\%, 100\%$ of the data. For the unlabeled set $\mathcal{D}^2$, we create subsets with $0\%, 10\%, 25\%, 50\%, 100\%$ of the data. Note that we have five settings of unlabeled data since we include the case of training with no unlabeled data as well.

### E.3 LOC Implementation

For all experiments, we initialize with $10\%$ of the training data set. We consider $T = 1, 3, 5$ rounds and sweep a range of $V^*$. We provide a summary of parameters in Table 5.

For the experiments with $K = 1$, we model the data requirement PDF $f(q)$ in each round of the problem as follows. We first draw $B = 500$ bootstrap resamples of the current training statistics $\mathcal{R}$, where $\mathcal{R} = \{(rq_0/R, V(\mathcal{D}_{rq_0/R}))\}_{r=1}^R \cup \{(q_s, V(\mathcal{D}_{q_s}))\}_{s=1}^t$ contains all of the measured statistics up to the initial data set (e.g., for CIFAR-10, this includes performance with $2\%, 4\%, \cdots, 10\%$ of the data), and the previous collected data. The latter is obtained by calling our piecewise-linear ground truth approximation. For each bootstrap resample, we fit a power regression model $\hat{v}(q; \boldsymbol{\theta}) = \theta_0 q^{\theta_1} + \theta_2$ and solve for the estimated minimum data requirement. We then use our set of

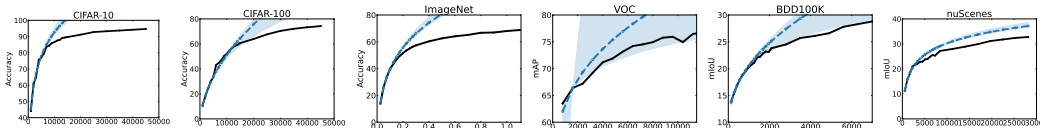

Figure 5: For a fixed seed, ground truth learning curves (black) and the estimated power law learning curves (blue) obtained via bootstrapping and ensembling. The shaded region represents the $95$ percentile of the ensemble and the dashed blue line represents the mean of the regression functions. The mean is consistently higher than the unknown ground truth, whereas the shaded region can at times cover it.

estimates to fit a Kernel Density Estimation (KDE) model after gridsearching for the best bandwidth parameter.

For the experiments with $K = 2$, we use the same above procedure but fit Gaussian Mixture Models (GMM) due to their having an easily computable CDF via the Gaussian $\text{erf}(\cdot)$, rather than numerically integrating the PDF. We grid-search over the number of mixture components for the GMM model.

We optimize over problems (4) and (7) using gradient descent techniques. Depending on the current state and data set, different hyperparameters perform better. As a result, we perform extensive hyperparameter tuning every time we need to solve the optimization problem. Here, we sweep over gradient descent with momentum and Adam with learning rates between $0.005$ to $500$.

We initialize each problem with $\mathbf{q}_t$ equal to the baseline regression solution and $\mathbf{q}_{t+s} = \mathbf{q}_t/(s + 1)$ for all $1 \le s \le T - t$. That is, we set the initial value for future collection amounts to be fractions of the initial value of the immediate amount of data to collect. We identified this initialization by manually inspecting the solutions found by LOC, consequently it improves the conditioning of the loss landscape relative to other random initialization schemes.

# F  Additional Numerical Results

This section contains expanded results of our numerical experiments and further ablations. Our key results include:

• In Appendix F.1, we evaluate the effectiveness of estimating $F(q)$ by plotting the estimated learning curves as well as the empirical histograms used to model the data requirement distribution.

• In Appendix F.2, we explore the sensitivity of our optimization algorithm to variations in the cost and penalty parameters. In all except one instance, LOC consistently maintains a low total cost and failure rate.

• In Appendix F.3, we explore the multi-variate LOC (i.e., $K = 2$) for problems where we have a small number of $T = 1, 3$ rounds. The baseline fails for almost all instances of $T = 1$, whereas LOC maintains a low failure rate.

• In Appendix F.4, we consider variants of LOC where we use different regression functions to estimate the data requirement distribution. Our optimization framework can be deployed on top of any regression function to reduce the failure rate.

## F.1  Estimating the Data Requirement Distribution $F(q)$

To estimate $F(q)$, we first create an ensemble of estimated learning curves, which we then invert to obtain an empirical distribution of estimated values for $D^*$. Figure 5 plots our bootstrap resampled estimated learning curves versus the ground truth performance for the first round of data collection when we have access to an initial $\mathcal{D}_{q_0}$ containing $10\%$ of the full data set. As noted in Mahmood et al. [2], the mean estimated learning curve diverges from the ground truth. However, by bootstrap resampling an ensemble of learning curves, we can cover the ground truth with some probability.

Figure 6 plots the empirical histograms of estimated $D^*$ as well as the estimated $F(q)$ obtained via KDE on CIFAR-10 with three different values for $V^*$. Although the mode of the estimated distribution is far from the ground truth $D^*$, the estimated distribution assigns some probability to the ground truth region. LOC optimizes over this estimated $F(q)$, which allows us to conservatively collect data and reduce the chances of failure.

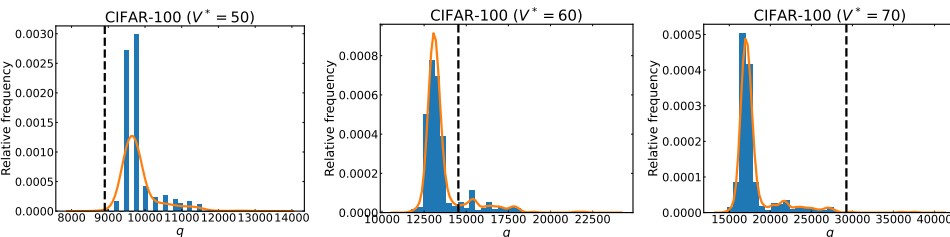

Figure 6: For a fixed seed, the histogram of estimates of $D^*$ from different bootstrapped models (blue bars), the estimated $F(q)$ (orange curve), and the ground truth $D^*$ (black dashed line). Each plot corresponds to a different $V^*$ for CIFAR-100 (see Figure 5 for the learning curve). With higher targets, regression (i.e., collecting the mean of the distribution) will lead to larger under-estimations.

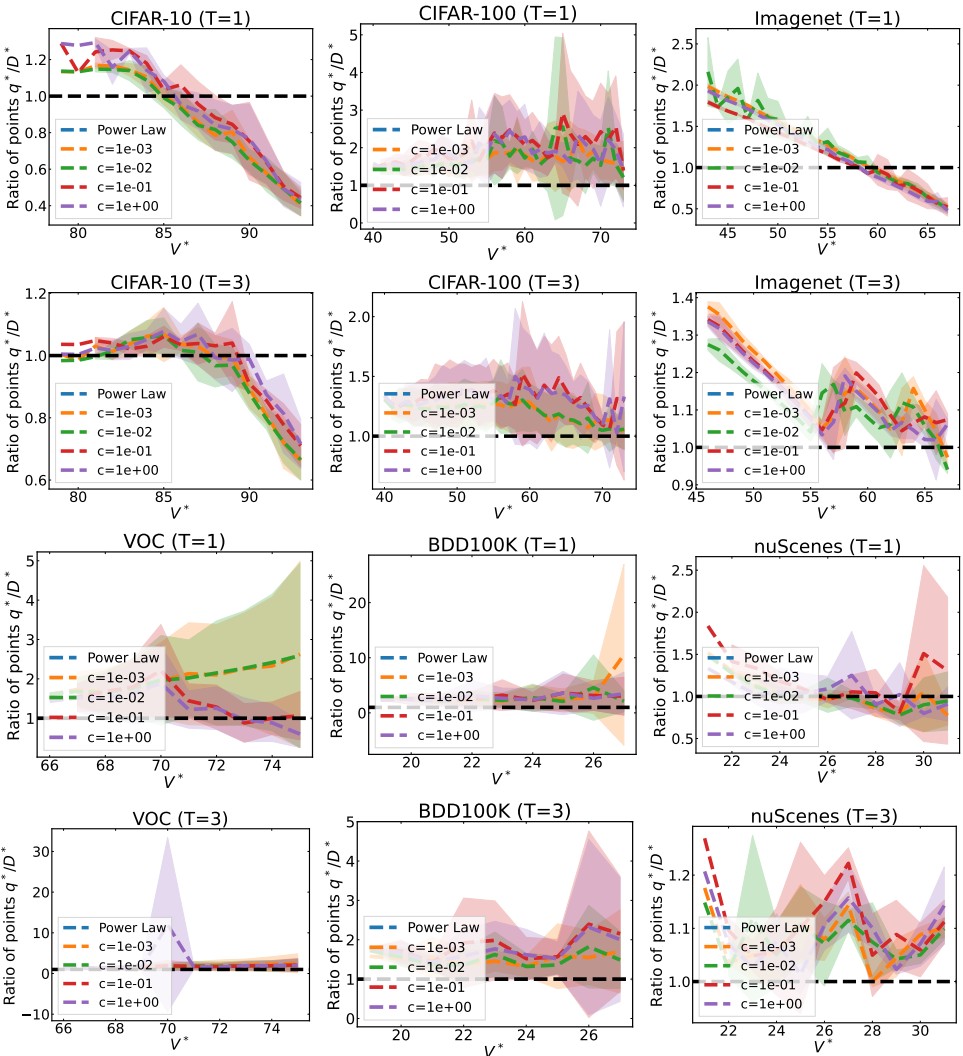

Figure 7: Mean $\pm$ standard deviation of the ratio of data collected $q_T^*/D^*$ for different $V^*$ when we sweep the cost parameter from $0.001$ to $1$ and fix $P = 10^7$. We show $T = 1, 3$ and refer to the main paper for $T = 5$. The dashed black line corresponds to collecting exactly the minimum data requirement.

## F.2  Robustness to the Cost and Penalty Parameters

Figure 7 expands the cost parameter sweep from Figure 3 (Top row) to the settings of $T = 1, 3$. For nearly all settings, LOC remains stable to variations in the cost parameter. Nonetheless, careful parameter selection becomes important as $T$ decreases. This is due to the fact that for low costs, the

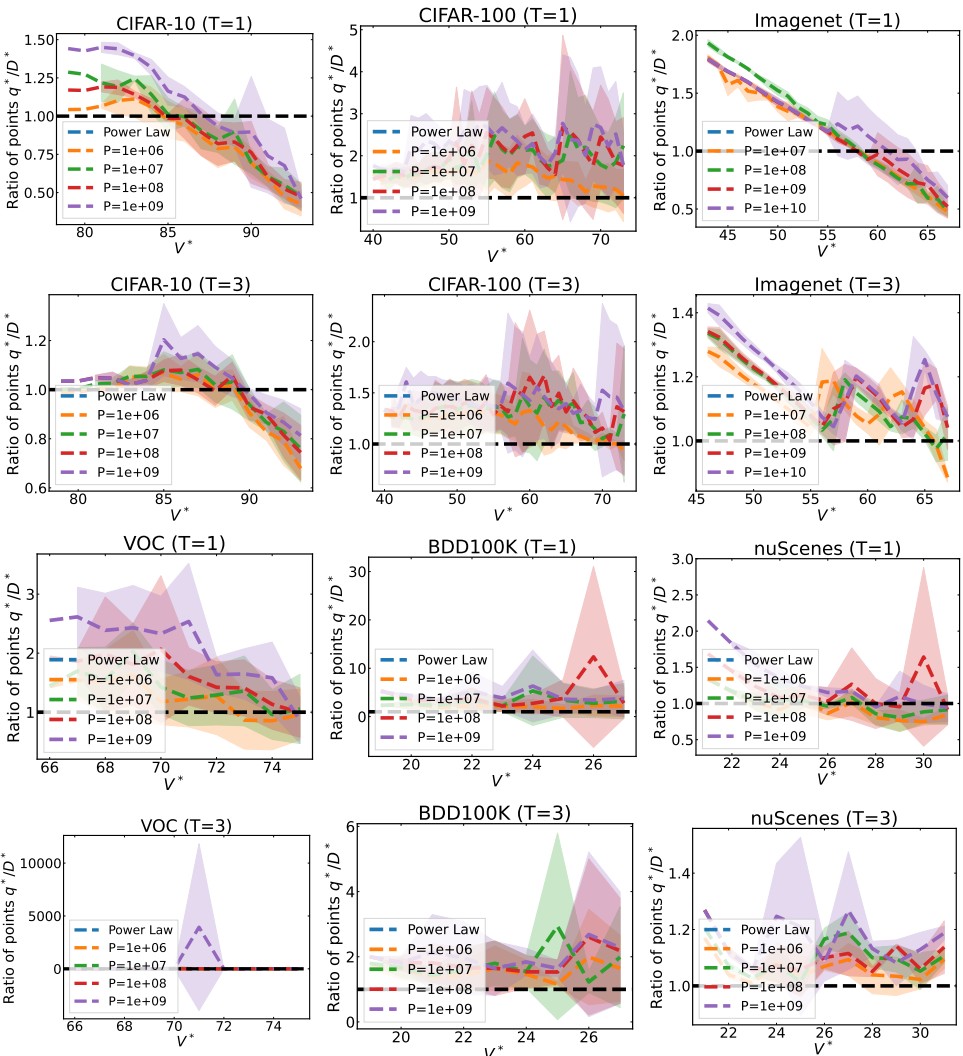

Figure 8: Mean ± std of the ratio of data collected $q_T^*/D^*$ for different $V^*$ when we sweep the penalty parameter from $10^6$ to $10^9$ and fix $c = 1$. We show $T = 1, 3$ and refer to the main paper for $T = 5$. The dashed black line corresponds to collecting exactly the minimum data requirement.

total amount of data collected increases as $T$ decreases (e.g., $c = 0.001$ for BDD100K). Furthermore, Figure 8 expands the penalty parameter sweep from Figure 3 (Bottom row). Here, we observe similar properties to the cost parameter sweep.

Although LOC is relatively stable on all other data sets, our results demonstrate some extreme results for VOC, potentially due to noise in the simulation. For example in Figure 8, setting $P = 10^9$, $V^* = 71$, and $T = 3$ led to collecting $10,000$ times the minimum data requirement. Such a situation is unrealistic in a production-level implementation, since in a real implementation, we could impose further constraints onto problem (4), such as upper bounds on the total amount of data permissible.

### F.3  The Value of Optimization over Estimation when $K = 2$

Figure 9 and Figure 10 expand Figure 4 to $T = 1, 3$ rounds. The results validate the summary observations from Table 2 in that the baseline has considerably higher failure rates versus LOC. In particular for BDD100K at $T = 1$, the baseline fails consistently for four out of five random seeds. On the other hand, recall that LOC admits a higher cost ratio compared to the baseline when $T = 1$. We can observe now that this high cost ratio is due to the method incurring high cost for a few target $V^*$ values. This behavior is similar to the observation above on VOC with high penalties at $T = 3$.

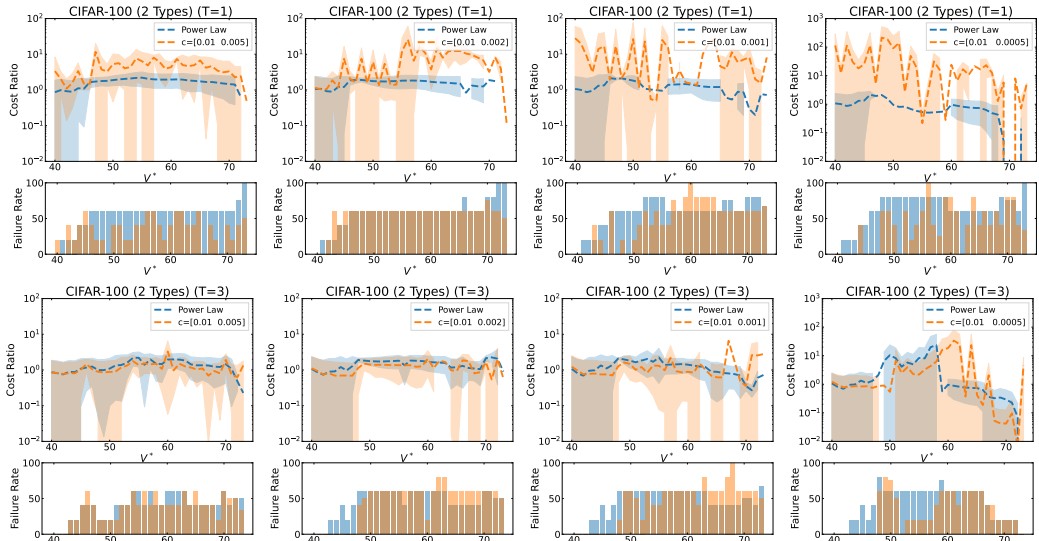

Figure 9: For experiments on CIFAR-100 with two data types, mean $\pm$ standard deviation over 5 seeds of the cost ratio $\mathbf{c}^\top(\mathbf{q}_T^* - \mathbf{q}_0)/\mathbf{c}^\top(\mathbf{D}^* - \mathbf{q}_0) - 1$ and failure rate for different $V$ after removing 99-th percentile outliers. We fix $c_0 = 1$ and $P = 10^{13}$. The rows correspond to $T = 1, 3$ (see the main paper for $T = 5$) and the columns correspond to $c_1 = c_0/2, c_0/5, c_0/10, c_0/20$.

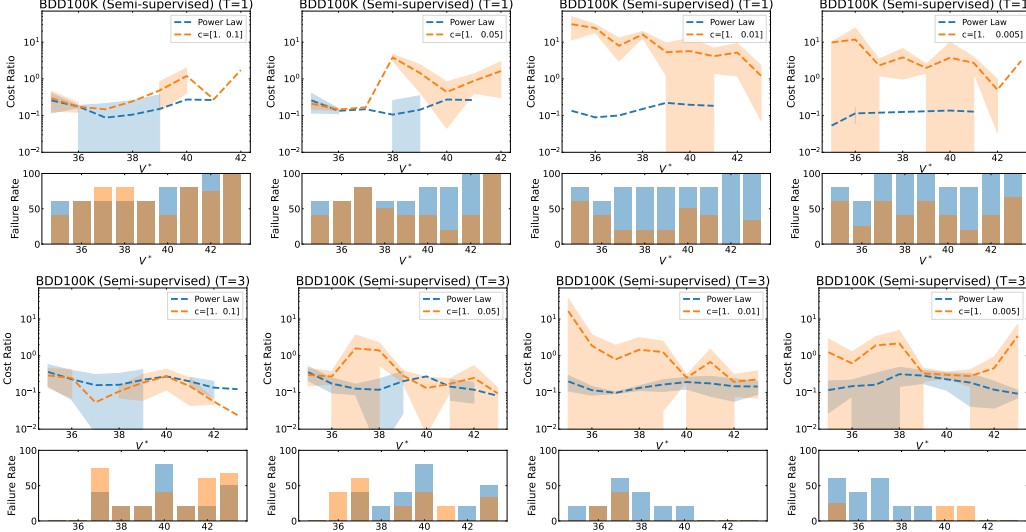

Figure 10: For experiments on BDD100K with two data types, mean $\pm$ standard deviation over 5 seeds of the cost ratio $\mathbf{c}^\top(\mathbf{q}_T^* - \mathbf{q}_0)/\mathbf{c}^\top(\mathbf{D}^* - \mathbf{q}_0) - 1$ and failure rate for different $V$ after removing 99-th percentile outliers. We fix $c_0 = 1$ and $P = 10^{13}$. The rows correspond to $T = 1, 3$ (see the main paper for $T = 5$) and the columns correspond to $c_1 = c_0/2, c_0/5, c_0/10, c_0/20$.

## F.4 LOC with Alternative Regression Functions

Mahmood et al. [2] show that we can use other regression functions instead of the power law to estimate the data requirement. Moreover, some functions tend to consistently over- or under-estimate the requirement. LOC can be deployed on top of any such regression function, since the regression function is only used to generate bootstrap samples.

Table 6 highlights experiments on CIFAR-100 with three alternative regression functions that were used by Mahmood et al. [2]. For both functions, we observe the same trends seen in Table 1. That is, LOC reduces the failure rate down to approximately zero, at a marginal relative increase in cost.

Noting that Power Law Regression often leads to failure, Mahmood et al. [2] also propose a correction factor heuristic wherein they learn a parameter $\tau$ such that if the data collection problem requires a

| Regression Function | $T$ | Regression | | LOC | |
|---|---|---|---|---|---|
| | | Failure rate | Cost ratio | Failure rate | Cost ratio |
| Logarithmic $\hat{v}(q;\boldsymbol{\theta}) = \theta_0 \log(q + \theta_1) + \theta_2$ | 1 | 43% | 0.19 | **2%** | 1.17 |
| | 3 | 37% | 0.17 | **2%** | 0.54 |
| | 5 | 34% | 0.16 | **1%** | 0.39 |
| Arctan $\hat{v}(q;\boldsymbol{\theta}) = \frac{200}{\pi} \arctan(\theta_0 \frac{\pi}{2} q + \theta_1) + \theta_2$ | 1 | 23% | 3.31 | **0%** | 5.56 |
| | 3 | 15% | 3.01 | **0%** | 3.92 |
| | 5 | 12% | 2.90 | **0%** | 3.60 |
| Algebraic Root $\hat{v}(q;\boldsymbol{\theta}) = \frac{100q}{1+|\theta_0 q|^{\theta_1})^{1/\theta_1}} + \theta_2$ | 1 | 52% | 0.11 | **23%** | 0.81 |
| | 3 | 44% | 0.1 | **2%** | 0.87 |
| | 5 | 44% | 0.1 | **2%** | 0.54 |

Table 6: For experiments on CIFAR-100, average cost ratio $\mathbf{c}^{\mathsf{T}}(\mathbf{q}_T^* - \mathbf{q}_0)/\mathbf{c}^{\mathsf{T}}(\mathbf{D}^* - \mathbf{q}_0) - 1$ and failure rate measured over a range of $V^*$ and $T$. We fix $c = 1$ and $P = 10^7$. The best performing failure rate for each setting is bolded. The cost ratio is measured only for instances that achieve $V^*$. LOC consistently reduces the average failure rate, almost consistently down to $0\%$.

| | Data set | $T$ | Regression With Correction [2] | | LOC | |
|---|---|---|---|---|---|---|
| | | | Failure rate | Cost ratio | Failure rate | Cost ratio |
| Class. | CIFAR-100 | 1 | 14% | 0.94 | 4% | 0.99 |
| | | 3 | **1%** | 0.23 | 3% | 0.31 |
| | | 5 | **0%** | 0.17 | 2% | 0.19 |
| | Imagenet | 1 | **7%** | 1.03 | 37% | 0.49 |
| | | 3 | **0%** | 0.21 | 5% | 0.16 |
| | | 5 | **0%** | 0.14 | 2% | 0.10 |
| Seg. | BDD100K | 1 | **4%** | 4.03 | 12% | 2.03 |
| | | 3 | **0%** | 1.02 | **0%** | 0.72 |
| | | 5 | **0%** | 0.62 | **0%** | 0.35 |
| | nuScenes | 1 | **0%** | 27.2 | 52% | 0.16 |
| | | 3 | **0%** | 0.75 | **0%** | 0.09 |
| | | 5 | **0%** | 0.30 | **0%** | 0.04 |
| Det. | VOC | 1 | **0%** | 44.6 | 25% | 0.56 |
| | | 3 | **0%** | 7.02 | **0%** | 1.10 |
| | | 5 | **0%** | 3.98 | **0%** | 0.84 |

Table 7: Comparison against the correction factor-based Power Law Regression of Mahmood et al. [2] using the same setup as in Table 1. The best performing cost ratio is underlined and the best performing failure rate for each setting is bolded. Although the baseline is designed specifically to achieve low failure rates, LOC often can achieve competitive failure rates while reducing the cost ratios by an order of magnitude.

target performance $V^*$, we should instead aim to collect enough data to meet $V^* + \tau$. In order to learn this correction factor, we require a pre-existing data set upon which we can simulate a data collection policy. Mahmood et al. [2] suggest setting $\tau$ such that we can achieve the data requirement $V^*$ for any $V^*$ on the pre-existing data set, and then fixing this parameter for new data sets.

Table 7 compares LOC (i.e., repeating Table 1) with the Correction factor-based Power Law regression baseline of Mahmood et al. [2]. Following the original paper, we tune $\tau$ using CIFAR-10 and apply it on all other data sets. The correction factor is designed to minimize the failure rate and thus, achieves nearly $0\%$ failure rate for all settings, but often at high cost ratios. On the other hand, LOC achieves generally low failure rates and low cost ratios. Specifically, for $T = 3, 5$, we are competitive with the baseline on failure rates for most tasks while obtaining up to an order of magnitude decrease in costs. For $T = 1$, we typically admit higher failure rates; however for the segmentation and detection tasks, we obtain up multiple orders of magnitude lower costs. Finally, note that this baseline requires a similar prior task to be effective. For example, the baseline outperforms us on cost and failure rate both only on CIFAR-100, since it is tuned on CIFAR-10. On the other hand, LOC does not require this prior data set to be effective as evidence by its performance on non-classification tasks.