# OpenReview forum: "Optimizing Data Collection for Machine Learning"
_NeurIPS.cc/2022/Conference — NeurIPS 2022 Accept_

### Official Review · Reviewer_buMu · 2022-06-19

**Rating:** 5
**Confidence:** 4
**Soundness:** 3 good
**Presentation:** 3 good
**Contribution:** 3 good

**Summary:**

This paper proposes a framework (LOC) for optimizing data collection workflows in machine learning applications with capturing the uncertainty in estimating data requirements. Besides, the paper further extends the proposed framework to multiple data resources cases.  The results show that LOC greatly reduces the chances of failing to meet the target performance.

**Questions:**

1. Report the baselines as stated in Weakness 1.

2. In your contribution statements (line 50), "We propose the optimal data collection problem in machine learning",   what is the difference between "the optimal data collection problem" and the data collection problem in Mahmood et al.[2]. It seems that both of your problems are to find the minimum datasets required to meet the target performance V^{*}.

3. How to effectively choose an optimal cost c. In Table 2, it seems that the cost ratio is very sensitive to the cost c. e.g. When T=5 on CIFAR100 and c is changing from (0.01,0.0005) to (0.01, 0.005), the cost ratio decreases to 0.82 from 5.19.

**Limitations:**

The authors have discussed about the limitation and social impacts.

**Strengths And Weaknesses:**

Strengths
1. This paper is well written and easy to follow and the motivation is clear.

2. The idea is novel and the proposed LOC can be applied to multiple data resources case.

Weakness
1. Important baseline is missing. Mahmood et al.[2] shows that the correction factor can boost the performance of power-law regression. It is necessary to report the results of power-law regression with a correction factor. Besides, the conventional estimation approach with other regression functions performs better than Power-law on some datasets (Mahmood et al.[2]), e.g., Algebraic Root consistently performs better than Power Law. It would be better to add the results of the convention estimation approach with Algebraic Root, Arctan.

2. The proposed LOC tends to overestimate the data requirements in some cases. e.g. In Figure 2., LOC has a much higher Ratio of  q_{T}^{*}/D^{*} on CIFAR-100 and BDD100K.

3. Some descriptions might be inaccurate/mistakes. e.g.

(1) In the caption of Figure 2., "LOC consistently remains slightly above the black line" might not be accurate since there exists cases where the ratio of q_{T}^{*}/D^{*} is below 1 (in high V^{*}) for LOC.

(2) In line 149, "problem (6) can be optimized via gradient descent" ---> "problem (4) can be...."?

---

> ### Author Response · Authors · 2022-08-02
> **Thank you for the review**
>
> Thank you for the positive feedback and the minor comments, which we have revised. We have updated the paper and Appendix (with revisions in blue).
>
> **Baselines (see response to Reviewer MHTm):**
>  We have revised Appendix F.3 to include more experiments with other regression functions and the correction factor from [1]. We outperform the other regression functions since optimization can be incorporated on top of any function and better functions that get better $F(q)$ estimates will yield better optimization solutions. The new results also show that the correction factor is designed for low failure rates, but achieves this with high costs. For $T=3, 5$, LOC often achieves the same failure rates but with up to $10\times$ less costs. The correction factor is only competitive for CIFAR-100, because it is tuned using CIFAR-10, which reveals another advantage of LOC: we do not need a prior related data set to tune the method. The main paper focuses on power laws since they are the most common scaling laws in the literature.
>
> **Contribution:**
>  Our optimal data collection problem generalizes the previous problem of [1] by incorporating costs and multiple data types, i.e., $K > 1$, for which it is much harder to estimate data requirements. For example, given expensive labeled data and cheap unlabeled data, we are not necessarily collecting the total least amount of data since it may be better to collect a lot of the (cheaper) data. We agree that when there is a single data type, our problem formulation recovers [1] as a special case.
>
> **Choosing the cost $c$:**
>  The cost parameter is typically not tunable and we recommend that it reflect the application. For example, if it is $20\times$ harder or more expensive to collect data for some longtail classes (e.g., finding patients with a rare disease in a medical application), then we must model that cost.
>
> **LOC overestimates data requirements for CIFAR-100 and BDD100K:**
> Although LOC admits higher cost ratios, Table 1 shows that the baselines fail to meet the target performance for between 20% to 70% of instances, whereas LOC fails less than 12% and often less than 5% of instances. This is because our optimization model uses a penalty $P$ that is orders of magnitude larger than the optimal collection costs. If we are willing to accept higher failure rates at lower cost ratios, we can reduce this parameter (e.g., Figure 3). However, we believe that high failure rates are more detrimental than costs since failing to meet a performance target might mean that an ML model cannot be deployed, incurring significant workflow delays.
>
>
> **References**
>
> [1] Mahmood, Rafid, et al. "How Much More Data Do I Need? Estimating Requirements for Downstream Tasks." CVPR. 2022.

---

### Official Review · Reviewer_kQGA · 2022-07-11

**Rating:** 7
**Confidence:** 3
**Soundness:** 3 good
**Presentation:** 3 good
**Contribution:** 3 good

**Summary:**

Labeling data is an important task in machine learning. However, it can also be expensive, making it unproductive to "over-label" data. Conversely, collecting too little data can lead to poor model accuracy. This paper presents an approach to determine the "right" amount of data to label for a particular model and task in order to achieve a provided target accuracy. The proposed framework can extend to a multi-dimensional setup, where some particular data items might be more expensive to label.

**Questions:**

- How accurately can $F(q)$ be estimated in practice? Also, does this procedure make sense if multiple versions of the model are being trained on various subsets of the full training dataset (including the full dataset itself)? Why do you need multiple rounds of data collection given the CDF $F(q)$?
- What does this mean: "In order to model the distribution of $D*$ , we can take $B$ bootstrap resamples of $R$"?
- "To avoid repeatedly sampling data, re-training a model, and evaluating the score, each 241 simulation uses a piecewise-linear approximation of a ‘ground truth’ learning curve that returns 242 model performance as a function of data set size." --> is the model trained at the end of this simulation with all of the collected data $q_0 + q_1 + … + q_T$ to check if the final accuracy of the model is greater than $V*$?
- Figure 2: is $D*$ the same across different values of $T$ for a given $V*$ value and task?
- What do the cost ratios with respect to labeling the entire dataset look like? I ask because if it's not substantially more expensive to just label the entire dataset every time, then going through this entire process is perhaps not necessary? In general, the cost ratio metric is a bit hard to understand.
- Why are failure rates so different across tasks for a given $T$ value (e.g., $T=1$)?
- "after removing 99-th percentile outliers" --> what are the different trials here? Different $V*$ values? Something more than that?

**Limitations:**

The authors did not discuss technical limitations of their work. Some discussion of problem settings where the approach doesn't work would be useful. The authors do discuss both the positive and negative societal implications of their work, which is helpful.

**Strengths And Weaknesses:**

#### Strengths
- The paper addresses an important problem that I think is understudied.
- The paper is well written and clear.
- The approach is clean, and seems to work well compared to relevant baselines like directly using a power law distribution.

#### Weaknesses
- I would have liked to see more exposition on the takeaways from the results. For example, why do the cost ratios not monotonically decrease as $T$ increases? In general, the main metrics presented are a bit confusing to understand.
- I would also have liked to see how effective estimates of the cumulative density function $F(q)$ are in practice.

---

> ### Author Response · Authors · 2022-08-02
> **Thank you for the review**
>
> Thank you for the strong positive feedback. We have updated the paper and Appendix (with revisions in blue).
>
> **Why do cost ratios not monotonically decrease:** For some instances, the cost ratios may increase with $T$. The optimization problem is non-convex, since $F(q)$ is a non-convex function, meaning we may find locally (but not globally) optimal solutions. Furthermore, the estimated learning curves may be sensitive to perturbations in the data, leading to the occasional poor fit (see our response to removing outliers).
>
> **Accurately estimating $F(q)$:** In our new Appendix F.4, we plot estimated learning curves and estimated histograms of $F(q)$. Estimating $F(q)$ is inherently difficult, which is why regression approaches that rely only on estimation will over- or under-shoot. Our optimization problem is designed to mitigate the uncertainty in estimation.
>
> **Training on various subsets:** We do not train the model on different subsets of the full  data set! The estimation problem begins with a small initial training set $\mathcal{D}_{q_0}$, and we subsample from various subsets of this initial set to estimate $F(q)$. By optimizing over the estimated distribution, we determine how much data to collect, but if the estimation is poor, then we may realize that we under-collected and we will have to collect data for another round.
>
> **Taking bootstrap resamples:** To estimate $F(q)$ (i.e., the distribution of $D^*$), we require an ensemble of estimated learning curves. We first train our model with subsets of the initial $\mathcal{D}_{q_0}$ to create a data set of statistics $\mathcal{R} := \{ (|\mathcal{D}_i|, V(\mathcal{D}_i)) \}$. We then use Bootstrap to sample with replacement from the set of statistics $\mathcal{R}$, fit a power law function with the resamples, and then estimate $D^*$ by inverting the function for $V^*$. We repeat this process $B$ times to construct a histogram for $F(q)$.
>
> **Piecewise-linear ground truth:** We use the simulation protocol of [1]. Before starting a simulation, we first sweep over subsets of the entire data set to construct a “ground truth”   learning curve, which is an oracle that can reveal what the performance is for a given data set size. This oracle is only used to evaluate a data collection method; after each round $t$, the oracle returns whether a model trained with $q_t$ data points will exceed $V^*$. We hope this answers the question, but please let us know if further clarification is required.
>
> **Figure 2:** Yes, in our experiments, for a fixed seed, the true minimum amount of data for a given $V^*$ is the same regardless of $T$.
>
> **Cost ratios for labeling the entire data set:**  The cost ratio measures the ratio of total amount spent by an estimator over the minimum possible amount needed to meet the performance target, i.e., the oracle-suboptimality of an estimator with respect to problem (1).
> Evaluating the cost ratio with respect to the complete data set would not be useful, since in real applications we do not have a fixed-size data set a priori. Instead, we can choose to collect an arbitrary amount of data; note that costs do not consider only labelling, but also sampling/generating a data point from the wild. We hope this clarifies the metric and we are happy to discuss it further.
>
> **Different failure rates:** Failure rates depend on how well we estimate the learning curve, which varies significantly between tasks. Furthermore, $T=1$ is the most challenging, since we have only one shot to estimate and collect more data than $D^*$; this increases the variance.
>
> **Removing 99-percentile outliers:** Due to the larger dimensionality of $K=2$, estimating $F(q)$ becomes difficult. Given a training seed and $V^*$, we occasionally obtain a poor estimator that incorrectly estimates an extremely large data requirement (e.g., collecting 1,000,000 unlabeled points for BDD100K). These outliers make our valid results more difficult to interpret since they would be easy to identify in practice (e.g., if a method suggests an amount that disagrees with our common sense, we would not collect that much data), and so, we exclude outliers for both methods.
>
> **References**
>
> [1] Mahmood, Rafid, et al. "How Much More Data Do I Need? Estimating Requirements for Downstream Tasks." CVPR. 2022.

---

> > ### Comment · Reviewer_kQGA · 2022-08-09
> > **Response**
> >
> > Thanks for the response. It would be helpful to include some of these details in the final version of the paper to help make it more readable.

---

### Official Review · Reviewer_MHTm · 2022-07-22

**Rating:** 4
**Confidence:** 4
**Soundness:** 3 good
**Presentation:** 3 good
**Contribution:** 2 fair

**Summary:**

This paper provides an extension to a recently proposed problem: how many data to collect to meet the requirement in a specific task when using a specfic model. The authors add one more key step to the orginal algortihm, making its performance better according to the results reported in the paper, and also capable of dealing with multi-variate tasks. The main contributions are:
1. Propose a new algorithm to solve the data collection problem based on an existed approach;
2. Extend the problem to a multi-variate setting;
3. Theoretically and practically prove the method has better performance compared to the previous algorithm;


**Questions:**

See the previous section.

**Limitations:**

The authors has discussed the limitation of this paper, and it is well-organized. That is, the cost c and the penalty P need to be given by the user, which is not always available.

**Strengths And Weaknesses:**

The main strengths are:
1. The problem itself is interesting and of growing interest;
2. The proposed method is technically sound;
3. According to the experiments results, the proposed method outperformes its baseline method;
4. Mostly the paper is well-organized.

The main weaknesses are:
1. The 'Related work' section seems overlaps heavily with the baseline paper -- not sure whether that's legal for Neurips, checking with the ACs;
2. The motivation for the new algorithm doesn't look strong. The key step of the proposed algorithm is to add a 'bootstrap sampling' after getting the support set V, to further estimate the distribution of the real optimal D by training separate regression models. This step can be time-consuming, considering one has to get a certain amount of data points to meaningfully simulate a distribution. Also, the data used for multiple regressions are different subsets of the overall support set V, which makes it unreasonable since for regression models, one generally needs more data to train a more precise model, spliting the dataset and train serveral sub-regression models and use them all for predictions does not make sense;
3. The experiments are not convincing enough. As reported in the original paper, they tried four types of regression and the results shown indicate different tasks correspond to different types of regression function in order to get the best performance. However, in this paper the authors only discussed one of those four functions. Also for the choice of K. only K=1, 2 are discussed, which is too few;
4. Minor issue. For line 139, the third line of the equation, did you mean qT rather than qs for the second term?
5. This may not be a weakness, but why would one wants to use these kind of framework at all if he/she already has a support set and validation set to roughly predict the number of samples needed by training a regression model on it?

---

> ### Author Response · Authors · 2022-08-02
> **Thank you for the review**
>
> Thank you for the positive feedback and for identifying the minor issue (#4). We have updated the paper and Appendix (with revisions in blue).
>
> **Related work:** We have revised this section accordingly.
>
> **Motivating bootstrap sampling:** Our algorithm is actually very fast! We have a one-time cost of re-training the neural network with approximately 10 growing subsets of the available data. We then take bootstrap resamples of these 10 training statistics and fit simple (3-parameter) power law regression models. It takes at most fifteen minutes to fit 500 bootstrap models.
> Bootstrap resampling is also a standard technique in deep learning uncertainty quantification [1] as well as approximating distributions [2], which further motivates our approach.
>
> **Experiments with different regression functions (see response to Reviewer buMu):** We have expanded Appendix F.3, to include additional experiments comparing against the other functions from [3] to show that optimization improves over other regression approaches as well. Specifically, our optimization can be incorporated on top of any regression function, and better regression functions should yield better distribution estimates, which should ultimately improve the optimization even further. We focus on power laws in the main paper since they are the most common scaling laws in the literature.
>
> **Experiments with larger K:** We believe $K=2$ reveals the most immediately impactful applications of data with different costs, e.g., labeled vs unlabeled data, real vs. synthetic data. Furthermore, simulations for $K > 2$ are extremely computationally prohibitive, because to evaluate the cost ratio, we need a “ground truth” learning curve. Constructing the ground truth requires re-training a model with $O(M^K)$ different subsets of the full data set, where $M \gtrapprox 15$ describes the granularity of the ground truth curve. We expand this point in our revised Appendix E.2.
>
> **Why use this framework:** This framework supposes that we have access to a validation set but only a small training set that is insufficient to achieve the desired validation performance. As a result, we must forecast how much additional training data we will need, collect this additional data, and then evaluate the model to see if we have reached our desired performance. Our framework shows that we can make better forecasts through our optimization problem.
>
> **References**
>
> [1] Abdar, Moloud, et al. "A review of uncertainty quantification in deep learning: Techniques, applications and challenges." Information Fusion 76. 2021.
>
> [2] Hastie, Trevor, et al. The elements of statistical learning: data mining, inference, and prediction. Vol. 2. 2009.
>
> [3] Mahmood, Rafid, et al. "How Much More Data Do I Need? Estimating Requirements for Downstream Tasks." CVPR. 2022.

---

### Author Response · Authors · 2022-08-02
**Thank you to all the reviewers**

We thank the reviewers for their detailed comments and positive feedback on the paper. All three reviewers stated that the problem is interesting, important, and clearly motivated; the method is technically sound, clean, and novel; and that the paper is well-organized and well-written.

This paper proposes an optimization framework to determine how much data to collect when given collection costs, a desired model performance, a timeline to reach that performance, and a penalty for failing. We develop an algorithm to optimize over the uncertainty of neural scaling law estimators. Our framework generalizes to the novel setting of multiple data sources with different costs (e.g., expensive labeled and cheap unlabeled, or expensive real and cheap synthetic data). Finally, experiments on classification, detection, and segmentation tasks show that on average, we achieve desired performances while minimizing costs.

We have revised the paper (with edits in blue) to address reviewer feedback. The common question was on including the alternative regression functions and correction factor baselines from [1]. Our revised Appendix F.3 expands our experiments with alternative regression functions and introduces comparisons against the correction factor baseline. We outperform the alternate regression functions. Because optimization can be applied over any regression function, better regression functions that yield better distribution estimates can only improve optimization further. Moreover, although the correction factor baseline is hand-tuned to reduce failure rates for meeting targets, our optimization approach achieves competitive failure rates at up to $10\times$ less cost.

**References**

[1] Mahmood, Rafid, et al. "How Much More Data Do I Need? Estimating Requirements for Downstream Tasks." CVPR. 2022.

---

### Meta-Review · Area_Chair_QVaM · 2022-08-29

**Recommendation:** Accept
**Confidence:** Certain

**Metareview:**

The paper addresses a critical problem in the era of massive-data-set ML, which is how to estimate the size of data that needs to be collected to train a model of a given performance level.  This is not the first paper to propose the problem, but it does give improvements and extensions over previous work.

The reviewers reached clear consensus that the problem is important, that the paper is extremely clear, and that the methods appear to work well.  However, there was concern that the current paper was perhaps not a significant step forward over a previous recent paper "How Much More Data Do I Need? Estimating Requirements for Downstream Tasks".  In order to help resolve this question, I read both the current submission (and revision), but also went back to read the previous paper -- in addition, of course, to fully going through all reviews and author responses.

In the end, I think that there is enough new material here to justify publication.  The use of the bootstrap seems to give useful improvement and is computationally tractable, and the extension to the multivariate / multi-source case is interesting and important.  But equally importantly, in my opinion, is the fact that the empirical work is so thoroughly done and adds additional empirical grounding to a nascent-but-critical area of the field.  For these reasons, I am discounting to some degree the arguments from the "borderline reject" review and am recommending acceptance.

Lastly, note that an earlier version of the paper included text from the related work that was disturbingly similar to that of a previously published paper.  The authors have since revised the paper and removed the similar / near overlapping text, which is appropriate, but even so should consider this a strong warning to avoid submitting overlapping text in the future.

**Award:**

No

---

### Decision · Program_Chairs · 2022-09-14

Accept